# Beyond Reactivity: Measuring Proactive Problem Solving in LLM Agents

## Abstract

LLM-based agents are increasingly moving towards proactivity: rather than awaiting instruction, they exercise agency to anticipate user needs and solve them autonomously. However, evaluating proactivity is challenging; current benchmarks are constrained to localized context, limiting their ability to test reasoning across sources and longer time horizons. To address this gap, we present PROBE (**Pr**oactive **R**esolution **o**f **B**ottlenecks). PROBE decomposes proactivity as a pipeline of three core capabilities: (1) searching for unspecified issues, (2) identifying specific bottlenecks, and (3) executing appropriate resolutions. We apply PROBE to evaluate leading LLMs and popular agentic frameworks, showing that even state-of-the-art models struggle to solve this benchmark. Computing our consistent measurements across frontier LLMs and agents, we find that the best end-to-end performance of 40% is achieved by both GPT-5 and Claude Opus-4.1. Additionally, we demonstrate the relative capabilities of each model and analyze mutual failure modes. Our results highlight the current limitations of autonomous action in agentic systems, and expose promising future research directions.

## 1 Introduction

Agentic systems built on Large Language Models (LLMs) have made immense progress, delivering practical value across several real-world applications including coding (Yang et al., 2024; Agashe et al., 2025), computer use (Song et al., 2024), web navigation (Zheng et al., 2024; Zhang et al., 2025), and healthcare (Kim et al., 2024; Sellergren et al., 2025). Despite significant progress, the majority of the agentic systems today are *reactive* - they expect explicit instruction from a user prior to attempting a task (Yao et al., 2023). To transcend their function as tools, agents need to be *proactive*: anticipating user needs from continuous observation, suggesting candidate tasks to address these needs, and executing these tasks reliably.

Prior studies on proactive agents have explored agent proactivity in interacting with physical environments (Zhang et al., 2023), asking follow-up questions (Zhang et al., 2024) and perceiving immediate needs from a personalized environment ?Yang et al. (2025a). However, existing approaches compress evaluation into narrow, immediate temporal context, failing to capture insights that emerge only through longer-term analysis. For instance, proactive agents that look only at current context would not detect and take an appropriate action for a missed deadline from the past (as shown in Figure 1). To this end, we operationalize proactivity as a three-part construct. Given a set of priorities and a personalized user datastore, agents **search** across documents for user-relevant issues, **identify** the most pertinent ones (which we term *bottlenecks*), and **resolve** said issues by executing appropriate actions.

Constructing a real-world benchmark for proactivity is difficult since collecting long time horizon, multi-document user data raises privacy concerns and creates significant annotation overhead. Building on previous successes in the generation of synthetic datasets Nadas et al. (2025); Long et al. (2024); Butt et al. (2024), we construct a data generation agent to build our benchmark (we describe this in section 2). The resulting PROBE benchmark comprises of 1,000 diverse samples that challenge AI systems to proactively identify and resolve critical bottlenecks hidden within realistic workplace datastores (see Figure1). Our evaluation reveals a striking capability gap: even state-of-the-art LLMs and specialized agentic frameworks achieve no more than 40% success on this end-to-end task, highlighting the substantial challenges that remain in developing truly proactive AI

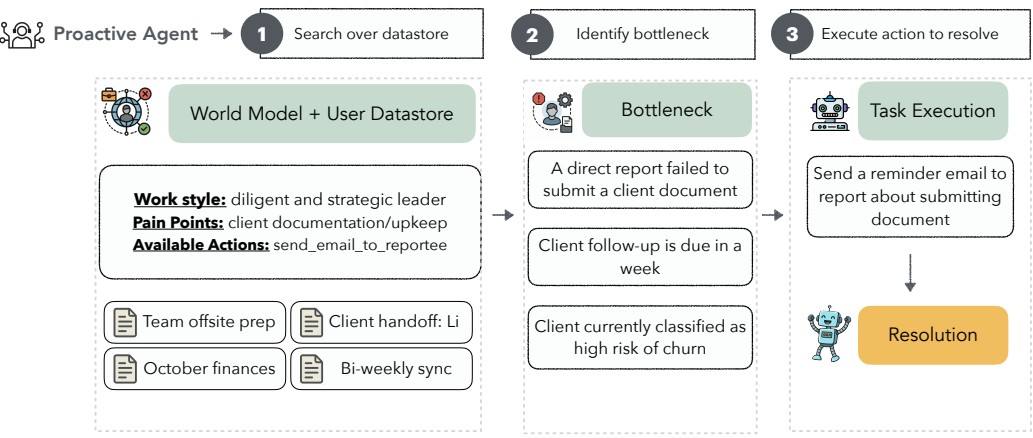

Figure 1: An end to end depiction of the PROBE task setup. The model (or agent) needs to use the world model to (i) search over a user datastore, (ii) identify the bottleneck and finally (iii) select the action to be executed. The model is evaluated across all tasks in this pipeline.

systems. From a model perspective, PROBE establishes a joint evaluation protocol for LLMs and agents, as shown in Figure 1. In summary, our contribution in this paper is threefold:

- We introduce PROBE - 1000 test samples that systematically evaluate proactive capabilities in AI systems through a unified framework, addressing a critical need for a realistic proactive benchmark.
- We conduct comprehensive evaluations across frontier closed-source and open-source models alongside leading agentic frameworks, revealing a fundamental capability ceiling: even the most advanced models achieve only 40% success on our end-to-end task.
- We present an in-depth analysis of common failure modes that uncovers the specific challenges associated with our benchmark and surfaces opportunities for future work.

## 2 METHODOLOGY

We develop a data generation pipeline to orchestrate our end-to-end workflow (described in Figure 2). Starting from real user personas, we build comprehensive world models that capture each simulated user's environment, goals, and constraints. These world models drive the creation of a datastore filled with personalized synthetic documents. While generating these documents, we ensure that several contain a pre-specified hidden obstacle (*bottleneck*). For each bottleneck, our pipeline generates several candidate actions, only one of which resolves the issue. This setup requires agents to demonstrate genuine proactivity: they must discover the bottleneck through exploration and identify the right fix among several plausible options. The following sections detail our problem formulation and pipeline components.

### 2.1 PROBLEM DEFINITION

**Setup and notation:** Consider a world-model $W$ of a user, let $D$ be a finite "universal" set of documents that constitutes the user's datastore, which is a collection of all of user's accessible documents, and let $b \in B$ denote a fixed *bottleneck*. We define a bottleneck as an issue that is critically important to the individual, actionable, and identifiable through a finite set of documents. For a given $b$, the rest of the documents that do not pertain to this bottleneck are considered *distractors* with respect to the bottleneck.

We define the binary predicate

$$f(d) : \forall d \in D \to \{0, 1\} \text{ evaluated under } W, \qquad f(d) = \begin{cases} 1 & \text{if } d \text{ conveys the bottleneck } b, \\ 0 & \text{otherwise.} \end{cases}$$

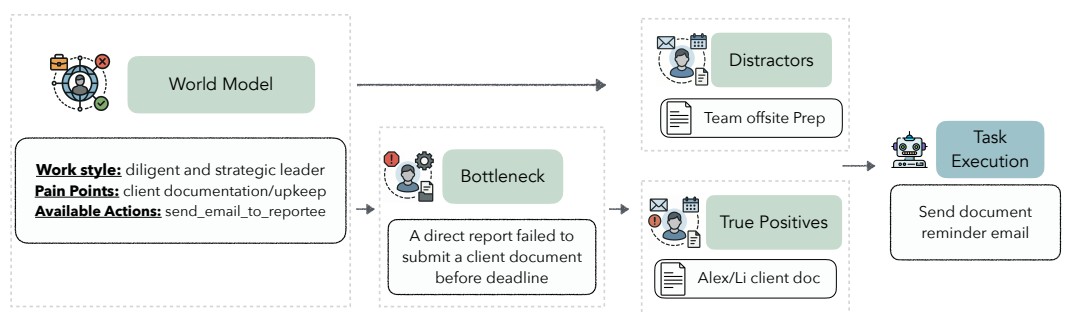

Figure 2: End-to-end proactive benchmark generation pipeline. A synthetic world model ($W$) is constructed in reference to a LinkedIn user profile, followed by generating a bottleneck ($b$) to resolve. True Positives ($T \subseteq D$) and distractors (other documents unrelated to the bottleneck, $K = D \setminus T$) are then constructed to frame the bottleneck prediction task. Finally, a task to execute is selected with parameters ( $\mathcal{P} = \{P_a\}_{a \in A}$ for action $a \in A$ ) to resolve the bottleneck. For any bottleneck, only one of the candidate tasks is a plausible resolution.

A document $d \in D$ is marked as a *true positive* (w.r.t. $b$) iff $f(d) = 1$, and a *distractor* iff $f(d) = 0$. For the current benchmark, we make the simplifying assumption that a true positive in a single data point contains evidence of only a single bottleneck.

**Sample generation:** Each instance of the benchmark is a tuple $S = (T, K, A, \mathcal{P}, b)$ where

- $T \subseteq D$ is the set of true positives for the sample with $|T| = t$,
- $K = D \setminus T$ is the set of distractors with $|K| = k$,
- $A$ is a finite set of available actions,
- $\mathcal{P} = \{P_a\}_{a \in A}$ assigns to each action $a \in A$ a parameter space $P_a$,
- $b$ is the bottleneck associated with this sample.

We make the assumption that the true-positive set $T$ is unique to this sample (i.e. different samples may not reuse the same $T$). The observed document set presented to the agent is $D := T \cup K$.

## 2.2 PROACTIVE TASK SETUP

An agent (LLM or agentic framework) receives an input $D$ and a set of $(a, P_a)$ tuples and must produce the tuple of outputs $\hat{O} = (\widehat{T}, \hat{b}, \hat{a}, \hat{p})$, where

- $\widehat{T} \subseteq D$ is the agent's predicted set of true positives,
- $\hat{b}$ is the agent's prediction for the bottleneck,
- $\hat{a} \in A$ is the selected action,
- $\hat{p} \in P_{\hat{a}}$ are the selected parameters for $\hat{a}$.

## 2.3 DATA GENERATION SETUP

We design a data generation pipeline that scales robustly in both context size and difficulty. Starting from real-world professional profiles, it constructs comprehensive synthetic world-models that mirror realistic workplace scenarios. Figure 2 illustrates the complete pipeline, which comprises four key components[1]:

**World Model Construction:** We leverage the dataset constructed by Ayoobi et al. (2023) to extract basic personas from real-world LinkedIn profiles. Each persona captures high-level professional information including current workplace, role description, and a professional summary.

---

[1] all prompts for individual data generation modules are shown in appendix C

From these personas, we synthetically construct comprehensive world models that encode:

- Workplace hierarchy and relationship context
- Work patterns and communication styles
- Available action space $A$ with corresponding parameter spaces $\mathcal{P}$
- Pain points and operational constraints

For instance, given a senior account manager with 20 years of client-facing experience as shown in Figure 2, the world model might identify "client documentation upkeep" as a pain point, while also modeling specific client relationships and their respective engagement contexts.

**Bottleneck Generation:** Using the contextualized world model, we generate bottleneck $b$: a persona-relevant, actionable user-need that satisfies our formal definition (see Section 2). Each bottleneck $b$ is designed to be identifiable through evidence $T$ in the document set $D$ and resolvable through exactly one action $a \in A$.

**User Datastore:** For each sample $S$, we construct the document set $D = T \cup K$. The **True positives** $T$ - documents where $f(d) = 1$ - collectively provide sufficient evidence to identify bottleneck $b$. **Distractors** $K$ are documents where $f(d) = 0$, introducing realistic noise with respect to the bottleneck. In our current datastore setup, all the generated documents are either emails, calendar events, or text documents, as exemplified in Figures 1 and 2. All generated documents are personalized, as generations are contextualized by the LinkedIn persona and world model.

To mirror real-world complexity, we employ two key design principles: (i) **Evidence distribution**: We often distribute evidence for $b$ across multiple documents in $T$, requiring agents to synthesize information from $t$ different sources. (ii) **Contextual noise**: We generate $k$ distractor documents of comparable length and professional relevance, ensuring bottleneck identification requires careful analysis rather than superficial pattern matching.

**Task Execution :** Finally, we construct the action set $A$ and parameter space for each action $a$, defined as $\mathcal{P} = \{P_a\}_{a \in A}$ such that:

- Exactly one action $a^* \in A$ effectively resolves bottleneck $b$
- Each action $a$ has a set of parameters $p \in P_a$ specified to resolve the bottleneck.
- Alternative actions represent potentially plausible but suboptimal interventions

For example, given a bottleneck about missing documentation, the optimal action $a^*$ might be *"send reminder email to direct report"* with parameters specifying the recipient, urgency level, and document details. The action set $A$ may include plausible alternatives such as "escalate to management" or "rewrite document", forcing agents to reason about the most effective intervention. All actions available for a bottleneck are populated as "available actions" in the user's world model.

## 2.4 PROBE - BENCHMARK FOR PROACTIVITY EVALUATION

We use the setup outlined above to generate the final dataset using GPT-4.1 as our primary model for generation. We generate a total of 1000 data points generated from 235 unique personas (described in 2.3). The full dataset stats are shown in Table 1[2]. We also provide a listing of bottleneck categories in Figure 3, as well as a complete world model-bottleneck combination in Appendix B.

|         | min   | max    | mean      | std     |
|---------|-------|--------|-----------|---------|
| Tokens  | 96294 | 122098 | 107,640.5 | 4,676.5 |
| Actions | 24    | 27     | 25.27     | 0.51    |
| Docs    | 70    | 81     | 79.3      | 3.7     |

Table 1: Dataset statistics. We show the statistics across number of tokens, actions and documents (true positives + distractors) for each instance.

To ensure the faithfulness of the dataset, we adapt a modified approach from Jiang et al. (2025) by performing multiple rounds of filtering via an adversarial agent (GPT-5) on a small sample set (of 5 data points). After each round of data generation, we asked the adversarial agent to identify and exploit patterns in the data, preferring

---

[2] token counts measured using tiktoken https://platform.openai.com/tokenizer

artifact-based solutions over independent reasoning. The final set of data points was generated only once no sample from our pipeline was solvable with these artifacts alone.

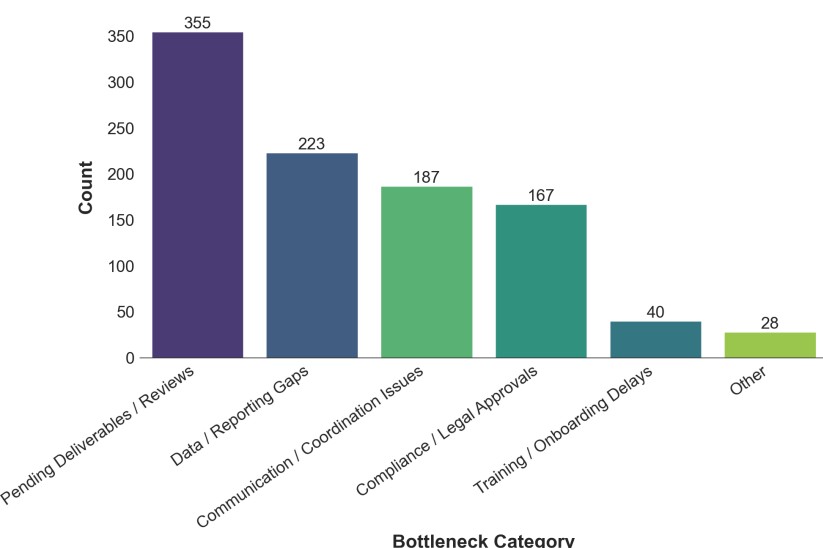

Figure 3: Distribution of bottleneck categories in the PROBE benchmark. The 1,000 synthetic test instances span six distinct workplace scenarios, ranging from pending deliverables and reviews to training gaps and communication breakdowns. These categories reflect realistic organizational challenges that proactive agents must identify and resolve through multi-document reasoning.

**Human Evaluation:** To establish human performance benchmarks and validate task difficulty, we conducted a 4-hour annotation study with three annotators, all holding at least a master's degree. Each annotator received identical instructions to those given to LLM systems and completed as many samples as possible within the specified time limit. Additionally, the annotators were instructed to judge if samples were realistic and feasible. To assess this, we asked the following questions: (i) Were all the documents you read realistic documents that you may see in a real workplace? and (ii) Were the actions you read realistic actions that you could see being used to resolve bottlenecks in a real workplace setting?

| Annotator ID | Search F1 % | Bottleneck Ident -tification Score (%) | Task Selection Score (%) | Entries Annotated | Samples hour | Realistic Artifacts |
|---|---|---|---|---|---|---|
| 1 | 26.79 | 0.00 | 0.00 | 13 | 3.25 | ✓ |
| 2 | 45.24 | 0.00 | 14.90 | 7 | 1.75 | ✓ |
| 3 | 20.37 | 0.00 | 8.33 | 6 | 1.50 | ✓ |
| Avg | 30.28 | 0.00 | 5.93 | 8.67 | 2.17 | ✓ |

Table 2: Annotation numbers showing that humans could not accomplish our task successfully. Annotators on average retrieved 30%, never identified the bottleneck, and selected the correct task at just above random chance, labeling about 2 samples per hour.

Across 12 total annotator-hours, only 26 samples were successfully completed at an average throughput of 2.17 samples per hour per annotator. The annotation experiment showed the substantial cognitive load required for bottleneck identification across multiple documents, and the time-intensive nature of synthesizing evidence and selecting appropriate actions[3]. For the annotators who answered "yes" to the two questions on judging artifacts, we record a green check mark in the "Realistic Artifacts" column of our results table. The annotation results for all three annotators shown in table 2. All human annotations were evaluated using the same metrics setup as described in section 3.

---

[3]Annotators found no sample to be unrealistic, and gave feedback that the task was very challenging

## 2.5 TASK CLARITY AND DATA QUALITY

To ensure task validity and data quality despite very poor human results, we conducted a controlled ablation study to disentangle task ambiguity from context-length difficulty. We hypothesized that the low human performance in the original study stemmed from the cognitive load of reasoning over massive context (107,640 tokens) rather than poor benchmark design. To test this, we created a "Short-Context" set by removing distractors and summarizing true positives, reducing the input to $\approx 1,000$ tokens.

| Annotator ID | Identification Score | Task Execution Score | Samples Annotated |
|---|---|---|---|
| 1 | 0.86 | 0.88 | 25 |
| 2 | 0.46 | 0.56 | 25 |
| 3 | 0.82 | 0.56 | 25 |
| **Average** | **0.71** | **0.67** | **Total: 75** |

Table 3: Human evaluation results after removing distractors and summarizing true positives. The substantial improvement in annotator performance indicates that the main challenge of the task stems from context length.

As shown in Table 3, human performance improves drastically under these conditions. We also record substantial inter-annotator agreement (Fleiss' $\kappa = 0.714$), providing evidence that the task is well defined: the difficulty for humans stems primarily from the challenge of long-context reasoning.

## 3 EVALUATION

### 3.1 METRICS

**Search:** This evaluation measures how well an agent retrieves the relevant documents that are required for identifying the bottleneck precisely. We measure the agent's retrieved document $\widehat{T}$ against the gold bottlenecks $T$ using standard precision, recall and F1 metrics.

**Bottleneck Identification:** Since this task uses a natural language output, we use the LLM-as-a-judge (Zheng et al., 2023) framework to evaluate whether the LLM identified the bottleneck correctly. We split this evaluation into two subtasks: (i) identifying essential details (who is the blocker, what is the task, root cause, etc.) and (ii) identifying non-essential details (system/tool names, processes to follow, scope of impact, etc.). The scoring rubric is as follows:

$$\text{Score} = \begin{cases} 1.0, & \text{if all essential and all non-essential details are accurate,} \\ 0.5, & \text{if all essential details are accurate but some non-essential details are incorrect,} \\ 0.0, & \text{if any essential detail is wrong or missing.} \end{cases}$$

**Task Execution:** Our scoring combines exact-match accuracy for the gold action label (assigning 0 if incorrect) with LLM-as-a-judge evaluation of parameter quality when the action is correctly identified. The rubric follows:

$$\text{Score} = \begin{cases} 1.0, & \text{if action predicted is correct and all critical parameters are present,} \\ 0.5, & \text{if action predicted is correct and most critical parameters are present,} \\ 0.0, & \text{action wrong or if critical parameters are missing.} \end{cases}$$

We provide both LLM-as-a-judge prompts in Appendix C.6.

### 3.2 LLM-AS-A-JUDGE

To validate our use of LLM-as-a-judge, we conducted a measurement study with 100 uniformly sampled GPT-4.1 prediction-output pairs. Two human annotators independently evaluated bottleneck identification and parameter judgments for each sample. We achieved 85% inter-annotator agreement and 81% human-LLM agreement across 100 total annotations, supporting our decision to use LLM-based scoring.

| | Search | | | Bottleneck Identification | Task Execution |
|---|---|---|---|---|---|
| | P | R | F1 | Score | Score |
| GPT-5 | **0.73** | **0.59** | **0.65** | 0.42 | **0.40** |
| Claude Opus 4.1 | 0.68 | 0.41 | 0.51 | **0.43** | **0.40** |
| Claude Sonnet 4 | 0.66 | 0.37 | 0.47 | **0.43** | 0.36 |
| GPT-4.1 | 0.60 | 0.38 | 0.46 | 0.42 | 0.38 |
| GPT-4.1-mini | 0.18 | 0.20 | 0.19 | 0.42 | 0.20 |
| DeepSeek-R1 | 0.49 | 0.23 | 0.29 | 0.04 | 0.19 |
| Kimi K-2 | 0.20 | 0.17 | 0.18 | 0.40 | 0.18 |
| GPT-OSS-120b | 0.27 | 0.10 | 0.13 | 0.35 | 0.11 |
| GPT-OSS-20b | 0.05 | 0.03 | 0.04 | 0.26 | 0.05 |

Table 4: Comparative results across several frontier closed and open source models. GPT-5 and Claude Opus-4.1 show the best performance compared to the rest. We also show evaluation across search, bottleneck identification and task execution to show the relative strengths and weaknesses across models. Note: We found GPT-5-mini performance to be close to GPT-5 performance across the board, hence its results are removed for brevity.

## 3.3 BASELINES

**Models**: We evaluate our benchmark against several frontier closed source models including OpenAI GPT-5 (OpenAI, 2025), GPT-5-mini (OpenAI, 2025c), GPT-4.1 (OpenAI, 2025a), GPT-4.1-mini (OpenAI, 2025b), Claude 4.1 Opus (Anthropic, 2025b), Claude 4 Sonnet (Anthropic, 2025a) and the best-performing open-source models including Kimi-K2 (Team et al., 2025) and DeepSeek-R1 (DeepSeek-AI et al., 2025). Among other open-source models, we test OpenAI GPT OSS (OpenAI, 2025) at both 120B and 20B scales[4].

**Agentic Frameworks**: We evaluate three leading agentic frameworks: ReACT (Yao et al., 2023), Reflexion-Agent (Shinn et al., 2023), and ReWOO (Xu et al., 2024). Since these frameworks target different problem domains, we adapted each for bottleneck resolution. Our modifications include minor prompt changes, structured outputs, and two retrieval tools: embedding based search and SQL queries. Each framework takes a distinct approach. ReACT cycles between reasoning and retrieval, progressively building context until it converges on an action. Reflexion learns from its failures: it runs multiple trials of document retrieval, analysis, and action selection, using LLM-based reflection to improve after each unsuccessful attempt. ReWOO, in contrast, precoordinates the process. After constructing a structured plan, it dispatches specialized workers for search and reasoning tasks, then synthesizes their findings to pinpoint bottlenecks and select interventions. We did not include any pre-existing proactive agent frameworks (Lu et al., 2024; Yang et al., 2025b) as baselines, as current systems are designed for specialized domains (conversational agents, UI navigation, embodied robotics). Resultantly, the distinct input modalities and task objectives of these frameworks do not trivially transfer to our workflow. All agentic frameworks use GPT-5-mini as the underlying base model. We provide more details in appendix A.

## 3.4 MODEL COMPARISONS

**Frontier Models Pull Ahead:** The gap between top models in our benchmark (GPT-5, GPT-4.1, Claude Opus, Claude Sonnet) and the rest of the models is significant. GPT-4.1-mini reaches 0.42 on Bottleneck Identification but only 0.20 on Task Execution, achieving just half the success rate of the best performing models. The other models fare poorly in retrieval and cascade these errors downstream (e.g., GPT-OSS-120b: 0.13 F1 Search, 0.11 Task Execution), underscoring the difficulty of end-to-end bottleneck resolution without strong evidence acquisition.

**Frontier models are stronger across the pipeline, but not uniformly:** GPT-5 achieves the best search performance with an F1 of 0.65 and the highest task execution score of 0.40, indicating stronger end-to-end capacity to find the right documents and translate bottleneck identification into an actionable plan. Claude Opus 4.1 and Claude Sonnet 4 achieve the best bottleneck identification

---

[4]We could not test Gemini-2.5 due to rate-limiting issues

| | Search | | | Bottleneck Identification | Task Execution |
|---|---|---|---|---|---|
| | P | R | F1 | Score | Score |
| ReACT(Yao et al., 2023) | 0.08 | 0.37 | 0.12 | 0.02 | 0.06 |
| Reflexion(Shinn et al., 2023) | 0.18 | 0.11 | 0.13 | 0.02 | 0.05 |
| ReWOO(Xu et al., 2024) | 0.27 | 0.24 | 0.25 | 0.01 | 0.11 |

Table 5: We show comparison across multiple agent frameworks. In our initial experiments, they significantly lag behind using LLMs out-of-the-box for this task, likely as a result of using retrieval tools instead of loading data into context.

score of 0.43 while having a slightly lower search score. This may suggest that Claude models have a slight advantage in reasoning capabilities for this task, which can offset potential shortcomings in search. Table 4 reveals that frontier models exhibit a variety of strengths: GPT-5 dominates search while Claude models lead in identification. Since achieving strong end-to-end performance requires balanced capabilities across all three dimensions, neither of these models scores well on the end-to-end task. Notably, DeepSeek-R1 presents an interesting anomaly: it performs competitively on search and execution metrics but substantially underperforms on bottleneck identification. On deeper inspection, we found that DeepSeek-R1 consistently used generic descriptions of bottlenecks instead of specific details, leading to reduced performance in bottleneck identification.

**Retrieval remains challenging:** Our analysis reveals a clear performance hierarchy among models. Frontier models (GPT and Claude series) significantly outperform others (Kimi K-2, DeepSeek-R1, GPT-OSS series), with all models showing higher precision than recall - for example, GPT-5 achieves 0.73 precision but only 0.59 recall. This pattern suggests models retrieve conservatively, struggling to some extent to retrieve all relevant pieces of information needed to identify the bottleneck. This is especially pronounced in smaller Language Models, which seem to massively under-retrieve relevant documents and resultantly struggle with any sort of task selection.

**Shortcutting helps overcome search difficulties, but not much:** Among the top-performing models (GPT-5, GPT-4.1, Claude Opus, Claude Sonnet), some compensate for weaker retrieval with stronger free-form reasoning during Bottleneck Identification. This yields competitive identification scores without a corresponding improvement in task execution. The gap highlights that being "right for the wrong reasons" does not translate into executable solutions. As the search space grows, this effect will degrade, reinforcing the need for faithful evidence use. We believe that the remaining head-room in this task will be based on faithful evidence use to identify bottlenecks and then resolve them correctly. The best bottleneck identification score reaches only 0.43, while the best task execution score is just 0.40. These low performance ceilings reveal significant gaps in current systems' ability to translate diagnoses into actionable solutions with complete parameters, particularly when retrieval is imperfect.

### 3.5 AGENT FRAMEWORKS COMPARISON

We evaluated agentic baselines using a constrained setup where agents were provided with a SQL store and embedding-based semantic search for document retrieval. Our rubric enforces three metrics across the pipeline: retrieval, bottleneck identification, and task execution. Across all tested agents, retrieval F1 scores ranged from 0.12 to 0.25, substantially below the frontier models in Table 5. This weak retrieval performance cascaded through the pipeline, resulting in Bottleneck Identification and Task Execution scores of $\leq 0.11$.

These results should be interpreted in context of the task structure. Standard agentic frameworks typically perform iterative, open-ended search over large external tools (e.g., web search, APIs). Our proactive task, however, requires agents to explore an unfamiliar datastore without a well-defined search target at the outset. This negates the benefit these frameworks attain from external tool use. Notably, all agentic frameworks are equitably evaluated: all access the data through SQL and Semantic Search.

## 4 Error Analysis

For this analysis, we use all failure cases of each model across the dataset. To understand where models struggle most, we analyze failure modes across three hierarchical categories: bottleneck identification, task selection, and parameter specification for the task that was selected (explained in metrics under section 3).

| Failure Mode | Claude Opus | Claude Sonnet | GPT-4.1 | GPT-5 | Kimi K-2 |
|---|---|---|---|---|---|
| *Identification Failures (% of identification errors)* | | | | | |
| Incorrect root cause[5] | 64.6% | 70.6% | 76.8% | 72.1% | 84.8% |
| Person attribution error | 46.9% | 57.9% | 61.3% | 60.8% | 78.0% |
| Missing/wrong deadline | 53.4% | 46.7% | 43.5% | 45.3% | 35.0% |
| *Function Selection Failures (% of action errors | identification success)* | | | | | |
| Wrong function selected | 9.7% | 9.5% | 9.2% | 10.6% | 10.9% |
| *Parameter Failures (% of action errors | function success)* | | | | | |
| Critical parameters missing | 66.2% | 75.4% | 79.5% | 65.2% | 71.7% |
| Incorrectly filled parameters | 45.4% | 36.1% | 35.3% | 45.6% | 44.3% |

Table 6: Failure mode breakdown across frontier models. Root cause identification emerges as the dominant failure mode, while function selection shows consistent competence across models.

**Root Cause Identification remains the primary challenge:** Incorrect root cause identification dominates across all models, averaging 73.8% of identification failures. This represents the single largest systematic weakness, with even the best-performing Claude Opus failing at root cause analysis in nearly two-thirds of identification errors. This suggests potential avenues to improve reasoning capabilities tailored to our proactivity task and may be a consequence of failures in the search stage.

**Interpersonal reasoning:** Interpersonal Reasoning - the ability of a model to identify and reason about the people involved in a bottleneck - remains challenging for models. All models substantially underperform in interpersonal dynamics (46.9%-78.0% failure rates), regardless of their performance on root cause identification. Even Claude Opus, the highest performing reasoning model in our benchmark, fails at interpersonal identification in nearly half of its errors, suggesting a difficulty in applying long context relationship information from the world model and documents.

**Action Selection and Parameter Prediction for actions :** Action selection and selection of parameters for the action remains independently challenging. GPT-5 and Claude Opus achieve better parameter coverage but higher error rates (45.6% and 45.4% incorrect), while GPT-4.1 and Claude Sonnet show the inverse pattern; more accurate specification (35.3% and 36.1% incorrect) but higher miss rates. Current models cannot simultaneously achieve higher coverage and precise parameter specification within the complex workplace scenarios seen in this benchmark.

## 5 Related Work

**Reactive vs. Proactive Agents:** Most LLM-based agent research has focused on reactive systems that respond to explicit user instructions. Key advances include planning approaches like ReAct (Yao et al., 2023), tool integration via Toolformer (Schick et al., 2023), and self-reflection mechanisms such as Reflexion (Shinn et al., 2023). While these expand agentic capabilities, they remain fundamentally reactive - dependent on explicit requests without capacity to anticipate user needs.

Emerging proactive agents face the challenge of anticipating latent user goals from partial observations and executing actions without explicit instruction. Recent work explores intent inference from behavioral patterns (Zhao et al., 2025), continuous insight generation from data streams (Yang et al., 2025b), and context-aware action generation (Shaikh et al., 2025). However, these systems lack systematic evaluation frameworks that assess end-to-end proactive capabilities.

---

[5]Incorrect root causes can signify either locating the possible cause and misidentifying the bottleneck or not identifying the possible cause at all.

**Agent Benchmarking**: Existing benchmarks predominantly evaluate reactive systems: SWE-bench (Jimenez et al., 2024) for software engineering; ToolBench (Qin et al., 2023) for API calling; and GAIA (Mialon et al., 2023) for multi-hop reasoning. Recent proactivity benchmarks like Proactive-VideoQA (Wang et al., 2025) and ProCIS (Samarinas & Zamani, 2024) begin addressing this gap but remain limited to conversational actions.

The Proactive Agent framework Lu et al. (2024) serves as our nearest counterpart, contributing a benchmark for annotated proactive suggestions in coding, writing, and everyday tasks. We build upon this work by benchmarking proactivity in long-horizon and multi-document datastores, incorporating search to uncover and address opportunities for assistance.

The above-mentioned works do not decompose proactivity into constituent capabilities or test comprehensive task execution across a multitude of sources and extended time horizons, motivating the systematic approach found in PROBE .

## 6 CONCLUSION

In this work we propose PROBE : a benchmark designed to test proactivity by having agents search over a personal datastore, identify bottlenecks without prompting, and resolve them. We evaluate leading LLMs and modern agentic solutions on this benchmark, and discover that most solutions struggle greatly at all three stages. We also conduct an analysis of failure modes, illustrating the difficulty of our benchmark's subcomponents.

While the work shows the difficulty of proactive assistance, it still only encompasses a part of the challenge: the problems of building good world-models for individual users and figuring out when to act remain unsolved. We leave these challenges to future work, with the hope that ongoing research will work towards personalized, dynamic agents that can identify and resolve the type of bottlenecks found in our benchmark.

## 7 LIMITATIONS AND FUTURE WORK

While our work advances proactive agent evaluation, several limitations present opportunities for future research. First, we assume a fixed, non-evolving world model across the time dimension. In real-world proactivity settings, personalization is a more fundamental component and represents a complex challenge. User preferences and contexts evolve over time, requiring agents to adapt their understanding dynamically. Second, we assume that for a given state of information, bottlenecks are resolvable by a single action. Many real-world bottlenecks involve complex, multi-step workflows that require dynamic task execution, where each action modifies the agent's state. These multi-step scenarios introduce additional complexity beyond the scope of this paper.

These limitations suggest natural directions for future work: developing benchmarks that incorporate temporal dynamics and evolving user models, and extending evaluation frameworks to handle multi-step bottleneck resolution with interdependent actions. Addressing these challenges will be essential for advancing proactive agents going forward.

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

## A  APPENDIX A: BASELINE IMPLEMENTATIONS

We provide implementation details for the agentic framework baselines. Full source code is available at `https://github.com/anonymized`.

All baseline agents share a common set of document retrieval tools and error handling mechanisms. The `semantic_search` tool performs vector-based retrieval using `text-embedding-3-small` embeddings with configurable result limits, while the `sql_reader` tool executes structured SQLite queries over document metadata with schema validation. All agents incorporate robust JSON parsing with fallback mechanisms and graceful error recovery for malformed outputs. These shared components ensure consistent document accessibility and reliable structured output generation across baselines.

### A.1  REACT

The ReACT agent follows the canonical Thought-Action-Observation pattern to iteratively reason about potential bottlenecks and search documents before selecting an appropriate response. The agent operates in a turn-based loop where each iteration generates a thought about the current context, selecting an action to take, and processing the resulting observations. The agent leverages both data exploration tools to retrieve relevant documents and action execution tools dynamically loaded from the world model. Once the agent identifies a bottleneck through its exploration and reasoning process, it selects an appropriate action from the available options and terminates, returning the retrieved documents, bottleneck description, and chosen action. To ensure robustness, the implementation includes safeguards such as token usage management for long contexts, maximum turn limits to prevent infinite loops, and fallback strategies for cases where the agent fails to converge on a definitive solution.

## A.2 Reflexion

The Reflexion agent leverages a verbal reinforcement learning approach that operates through iterative trial-and-error with self-reflection. The agent follows a structured three-step workflow: first searching for relevant documents, then analyzing retrieved documents to identify workflow bottlenecks, and finally selecting appropriate actions to address the identified issues. What distinguishes this architecture is its reflection mechanism: when an attempt fails to meet a quality threshold (evaluated by an LLM-based scoring system), the agent generates verbal reflections on what went wrong and incorporates these learnings into subsequent trials. This creates a feedback loop where the agent progressively improves its document retrieval strategies, bottleneck identification accuracy, and action selection through accumulated reflections from previous failures. The system runs multiple trials until either achieving a successful result (score $\geq 0.8$) or exhausting the maximum number of attempts, making it particularly effective for complex productivity tasks that require iterative refinement and learning from mistakes.

## A.3 ReWOO

The ReWOO consists of a three-stage modular workflow for bottleneck identification and resolution. The architecture follows a Plan-Work-Solve paradigm where the system first generates a structured plan to gather evidence, executes that plan using specialized workers, and then synthesizes the evidence to identify bottlenecks and propose actions. The Planner component analyzes the user's world model to create a step-by-step evidence gathering strategy, storing intermediate results in variables ($\#E1$, $\#E2$, etc.). The Worker stage then executes this plan using three specialized tools: `semantic_search` and `sql_reader` for finding relevant documents and LLM reasoning for analysis. Finally, the Solver component reviews all gathered evidence to identify the most critical bottleneck pattern and select the appropriate action from the available options.

## B Appendix B: Example World Model and Bottleneck

This appendix provides a complete example of a world model and a bottleneck used in our benchmark. The example demonstrates the persona context, relationships, and available actions that the agent must reason over.

## B.1 World Model

**World Model Example**

The following JSON structure represents a complete world model instance from our benchmark dataset, illustrating the personalized level of detail and complexity agents must navigate in PROBE.

```
{
  "world_model": {
    "persona_id": "connor_smith",
    "persona_full_name": "Connor Smith",
    "persona_occupation": "Financial Services Lead Advisory, PwC UK",
    "persona_about": "A Business graduate with a strong interest in Finance currently working for PwC
      within the Corporate Finance practice. ACA exam qualified with ICAEW, all first time passes. A
      Manager in the Financial Services Lead Advisory team within PwC UK. The FSLA team combines deep
      industry knowledge and transaction experience to advise on all aspects of corporate finance M&A and
      , through the Portfolio Advisory Group, credit related transactions. The team leverages more than
      6,000 PwC people in the U.K. and 60,000 PwC people globally who work day in day out with financial
      services clients.",
    "relationships": [
      {
        "name": "Emily Patel",
        "type": "manager",
        "department": "Financial Services Lead Advisory",
        "relationship_context": "Direct line manager overseeing Connor's progression from Senior
      Associate to Manager, providing strategic direction on major M&A deals and regular feedback on
      client engagement style."
      },
      {
        "name": "James Thornton",
        "type": "colleague",
        "department": "Portfolio Advisory Group",
```

```
      "relationship_context": "Collaborate intensively on distressed credit portfolio transactions,
    often negotiating differing approaches to valuation methodologies but ultimately delivering
    cohesive client recommendations."
     },
     {
       "name": "Priya Malhotra",
       "type": "direct_report",
       "department": "Financial Services Lead Advisory",
       "relationship_context": "Recently promoted Senior Associate who Connor is mentoring through her
    first lead on a mid-market bank acquisition, balancing support with constructive critique."
     },
     {
       "name": "Olivia Green",
       "type": "client",
       "department": null,
       "relationship_context": "Senior Corporate Development Manager at a challenger bank, relies on
    Connor's team for buy-side advisory; the relationship is trusted but currently tense due to a
    delayed regulatory approval."
     },
     {
       "name": "Michael Zhang",
       "type": "stakeholder",
       "department": "Risk Assurance",
       "relationship_context": "Internal stakeholder who reviews risk elements of Connor's transactions
    ; their differing perspectives on risk appetite have led to constructive debates and enhanced deal
    structuring."
     },
     {
       "name": "Sophie Laurent",
       "type": "vendor",
       "department": null,
       "relationship_context": "Relationship manager at a third-party data analytics firm providing due
     diligence support, with whom Connor negotiates service level agreements and escalates data quality
     issues."
     },
     {
       "name": "Rebecca Yates",
       "type": "stakeholder",
       "department": "Compliance",
       "relationship_context": "Works closely with Connor to ensure all deal processes meet FCA
    regulations, occasionally challenging his team's interpretation of compliance requirements."
     },
     {
       "name": "TomÃąs Ortega",
       "type": "client",
       "department": null,
       "relationship_context": "Head of M&A at a European insurance group, Connor manages international
     communications and project delivery, navigating cultural nuances and tight deadlines."
     },
     {
       "name": "Grace Lin",
       "type": "mentee",
       "department": "Financial Services Lead Advisory",
       "relationship_context": "Graduate Analyst whom Connor is coaching on financial modeling; their
    regular sessions have fostered Grace's rapid development, though she sometimes struggles with
    client-facing confidence."
     },
     {
       "name": "Andrew Williams",
       "type": "cross_functional_partner",
       "department": "Technology Consulting",
       "relationship_context": "Collaborate on digital transformation due diligence for fintech targets
    , balancing differing priorities between operational efficiency and deal speed."
     },
     {
       "name": "Marta Kowalska",
       "type": "vendor",
       "department": null,
       "relationship_context": "Director at a legal advisory firm providing transaction legal due
    diligence; Connor manages the engagement and ensures alignment on documentation timetables."
     },
     {
       "name": "Ben Carter",
       "type": "direct_report",
       "department": "Financial Services Lead Advisory",
       "relationship_context": "Associate who handles initial deal screening and supports Connor on
    market research, recently needing extra guidance due to a challenging workload."
     },
     {
       "name": "Hannah Robinson",
       "type": "colleague",
       "department": "Valuations",
       "relationship_context": "Works with Connor on valuations for complex asset portfolios; their
    differing technical approaches often lead to productive, data-driven debates."
     },
     {
       "name": "Samir Ahmed",
       "type": "stakeholder",
       "department": "Human Capital",
       "relationship_context": "Partners with Connor's team for talent allocation on high-value deals;
    occasionally tensions arise during peak periods over resource prioritization."
     },
```

```
864
865        {
866          "name": "Julia Becker",
           "type": "client",
867          "department": null,
           "relationship_context": "Private equity partner who values Connor's candid market assessments,
868        but has recently challenged his team's fee structure on a major divestment mandate."
869        },
           {
870          "name": "Freddie White",
           "type": "mentee",
871          "department": "Financial Services Lead Advisory",
           "relationship_context": "Early-career team member Connor supports through the ICAEW
872        qualification process, providing both technical guidance and emotional support during exam periods
           ."
873        },
           {
874          "name": "Isabelle Dubois",
           "type": "cross_functional_partner",
875          "department": "Sustainability & ESG Advisory",
           "relationship_context": "Works with Connor to integrate ESG considerations into financial
876        advisory pitches, sometimes pushing Connor to adapt more progressive frameworks for skeptical
877        clients."
           }
878      ],
         "personal_context": {
879        "work_style": "Connor operates in a high-pressure, fast-paced environment, balancing multiple M&A
880        transaction pipelines with ongoing client advisory work. He prioritizes structured project
           management, but frequently adapts to shifting client demands and tight regulatory timelines. He
881        leverages both collaborative teamwork and independent analysis, often working across time zones and
            balancing strategic oversight with detailed financial diligence.",
882        "communication_preferences": [
           "Email for formal updates and transaction documentation",
883          "Video calls for deal negotiation and client relationship management",
           "Instant messaging (e.g., Microsoft Teams) for quick internal queries and urgent project
884        coordination",
           "Phone calls for time-sensitive decisions or when clarity is critical",
885          "Face-to-face meetings for complex client presentations and team alignment sessions"
         ],
886        "current_priorities": [
           "Managing multiple concurrent M&A transactions with overlapping deadlines",
887          "Coordinating due diligence efforts across PwC teams and external stakeholders",
888          "Ensuring regulatory compliance and risk mitigation within all live deals",
           "Developing junior team members while maintaining project delivery standards",
889          "Enhancing client relationships and identifying cross-sell opportunities",
890          "Integrating Portfolio Advisory Group insights into standard deal processes",
           "Meeting internal performance and billing targets while navigating resource constraints"
891      ],
         "pain_points": [
892          "Juggling conflicting deadlines across several high-stakes projects",
893          "Difficulty in accessing timely inputs from cross-functional or global teams",
           "Managing client expectations amid rapidly changing financial markets",
894          "Keeping up with evolving regulatory requirements and ensuring compliance",
895          "Limited time for proactive client development due to reactive workload",
           "Onboarding and mentoring new staff without compromising on project quality"
896        ]
       },
897      "available_actions": [
         {
898          "id": "conduct_peer_review_of_draft_report",
899          "type": "conduct_review",
           "description": "Carry out a peer review of a draft report to ensure quality and compliance
900        standards.",
           "constraints": [
901            "peer_reviewer_assigned",
902            "review_criteria_defined"
           ],
903          "params_schema": {
           "required": [
904              "document_id",
905              "reviewer",
             "review_feedback"
906          ]
           }
907        },
         {
908          "id": "create_briefing_note_on_sector_updates",
909          "type": "create_document",
           "description": "Develop a briefing note summarizing key sector updates for internal awareness.",
910          "constraints": [
           "sector_data_compiled",
911            "briefing_template_used"
912        ],
           "params_schema": {
913          "required": [
             "briefing_date",
914            "sector",
             "summary_points"
915          ]
           }
916      },
         {
917          "id": "review_and_approve_time_entries",
```

```
            "type": "update_task_status",
            "description": "Update the approval status of submitted time entries for the previous month.",
            "constraints": [
              "time_entries_submitted",
              "review_period_open"
            ],
            "params_schema": {
              "required": [
                "timesheet_id",
                "approval_status",
                "review_comments"
              ]
            }
          },
          {
            "id": "finalize_presentation_for_external_workshop",
            "type": "update_document",
            "description": "Make final adjustments to the presentation slides for an upcoming external
        workshop.",
            "constraints": [
              "slide_content_approved",
              "branding_guidelines_followed"
            ],
            "params_schema": {
              "required": [
                "document_id",
                "sections_to_edit",
                "visual_elements"
              ]
            }
          },
          {
            "id": "initiate_internal_knowledge_share",
            "type": "schedule_meeting",
            "description": "Arrange a knowledge-sharing session for the team to exchange best practices on
        due diligence processes.",
            "constraints": [
              "subject_matter_experts_available",
              "content_outline_prepared"
            ],
            "params_schema": {
              "required": [
                "attendees",
                "date",
                "time",
                "session_topic"
              ]
            }
          },
          {
            "id": "send_reminder_for_time_allocation_reconciliation",
            "type": "send_email",
            "description": "Send a reminder to the operations team to expedite reconciliation of time
        allocation data in the finance system.",
            "constraints": [
              "prior_request_sent",
              "pending_reconciliation"
            ],
            "params_schema": {
              "required": [
                "recipients",
                "subject",
                "reminder_message"
              ]
            }
          },
          {
            "id": "initiate_annual_training_enrollment",
            "type": "create_task",
            "description": "Create a task to enroll the team in mandatory annual training sessions.",
            "constraints": [
              "training_schedule_available",
              "team_member_list_current"
            ],
            "params_schema": {
              "required": [
                "training_type",
                "enrollment_deadline",
                "participant_list"
              ]
            }
          },
          {
            "id": "organize_quarterly_team_checkin",
            "type": "schedule_meeting",
            "description": "Schedule a quarterly meeting for the team to discuss ongoing initiatives and
        operational improvements.",
            "constraints": [
              "team_calendar_availability",
              "meeting_room_availability"
            ],
            "params_schema": {
```

```
972
973              "required": [
974                "attendees",
975                "date",
976                "time",
977                "agenda"
                 ]
               }
             },
978           {
979             "id": "submit_request_for_client_feedback",
980             "type": "request_approval",
               "description": "Submit a formal request for client feedback on recently delivered project
               outputs.",
981             "constraints": [
982                "project_deliverables_completed",
983                "client_point_of_contact_confirmed"
               ],
984             "params_schema": {
985                "required": [
986                  "client_id",
                     "project_id",
987                  "feedback_request_message"
                 ]
               }
             },
988           {
989             "id": "initiate_information_request_to_central_finance_team",
990             "type": "send_email",
               "description": "Send a follow-up email to the central finance group to request updated reference
               data required for current project models.",
991             "constraints": [
992                "previous_request_no_response",
993                "project_deadlines_approaching"
               ],
994             "params_schema": {
995                "required": [
                     "recipients",
996                  "subject",
                     "message_body"
                 ]
               }
             },
997           {
998             "id": "reply_to_team_query_on_policy_update",
999             "type": "reply_email",
               "description": "Reply to a team member's email regarding recent policy updates and clarify
1000          outstanding questions.",
1001            "constraints": [
1002               "policy_documentation_on_hand",
                   "clarification_needed"
1003          ],
1004            "params_schema": {
1005               "required": [
                     "thread_id",
1006                 "response_content"
                 ]
               }
             },
1007          {
1008            "id": "cancel_redundant_calendar_event",
1009            "type": "cancel_meeting",
               "description": "Cancel an outdated or redundant calendar event to reduce scheduling conflicts.",
1010            "constraints": [
1011               "event_no_longer_needed",
                   "invitees_notified"
1012          ],
1013            "params_schema": {
1014               "required": [
                     "meeting_id",
                     "cancellation_reason"
                 ]
               }
             },
1015          {
1016            "id": "reschedule_internal_alignment_meeting",
1017            "type": "reschedule_meeting",
               "description": "Move the internal alignment meeting to accommodate conflicting priorities.",
1018            "constraints": [
1019               "attendee_availability_checked",
                   "calendar_invite_updated"
1020          ],
1021            "params_schema": {
1022               "required": [
                     "meeting_id",
1023                 "new_date",
                     "new_time"
                 ]
               }
             },
1024          {
1025            "id": "prepare_meeting_agenda_for_upcoming_review",
               "type": "create_meeting_agenda",
```

```
1026
1027          "description": "Draft an agenda for an upcoming review meeting with stakeholders.",
1028          "constraints": [
               "meeting_objectives_defined",
1029           "stakeholder_input_collected"
              ],
1030          "params_schema": {
               "required": [
1031            "meeting_date",
1032            "agenda_items",
               "expected_outcomes"
1033           ]
              }
1034        },
           {
1035          "id": "assign_documentation_task_to_team_member",
1036          "type": "delegate_task",
             "description": "Delegate the task of documenting process flows to a designated team member.",
1037          "constraints": [
               "team_member_availability",
1038           "skill_relevance"
              ],
1039          "params_schema": {
               "required": [
1040            "assignee",
1041            "task_description",
               "due_date"
1042           ]
              }
1043        },
           {
1044          "id": "request_additional_resources_for_project",
1045          "type": "create_task",
             "description": "Open a task to formally request extra resources for a high-priority project due
1046    to increased workload.",
             "constraints": [
1047           "resource_justification_prepared",
               "project_timeline_affected"
1048         ],
             "params_schema": {
1049           "required": [
1050            "project_id",
               "resource_type",
1051            "rationale"
              ]
1052         }
           },
1053        {
1054          "id": "compile_weekly_market_summary_report",
             "type": "create_document",
1055          "description": "Prepare a weekly summary document highlighting recent market trends for internal
1056     distribution.",
             "constraints": [
1057           "market_data_access",
               "standard_formatting"
1058         ],
             "params_schema": {
1059           "required": [
1060            "report_date",
               "market_segments",
1061            "key_findings"
              ]
1062         }
           },
1063        {
1064          "id": "follow_up_on_pending_client_information",
             "type": "send_slack_message",
1065          "description": "Send a message to the client liaison to check on the status of outstanding
1066    information requests.",
             "constraints": [
1067           "previous_request_sent",
               "response_delay"
1068         ],
             "params_schema": {
1069           "required": [
1070            "recipient",
               "message_content"
1071           ]
             }
1072        },
           {
1073          "id": "send_status_update_on_deliverables",
1074          "type": "send_email",
             "description": "Send an email to project stakeholders with a status update on key deliverables
1075    and next steps.",
             "constraints": [
1076           "stakeholder_list_current",
               "deliverable_status_confirmed"
1077         ],
             "params_schema": {
1078           "required": [
1079            "recipients",
               "subject",
```

```
1080                    "update_content"
1081                  ]
1082                }
              },
1083              {
1084                "id": "make_call_to_confirm_external_deadlines",
1085                "type": "make_phone_call",
                  "description": "Place a call to confirm upcoming external deadlines for a strategic initiative
1086            .",
1087                "constraints": [
                    "contact_information_verified",
1088                  "relevant_documents_prepared"
1089                ],
                  "params_schema": {
1090                  "required": [
                      "contact_number",
1091                    "call_objective",
1092                    "call_notes"
                    ]
1093                }
1094              },
              {
1095                "id": "escalate_technical_support_ticket",
                  "type": "escalate_to_manager",
1096                "description": "Escalate an unresolved technical support ticket to the appropriate escalation
1097            channel for prioritized handling.",
                  "constraints": [
1098                  "ticket_open_exceeds_threshold",
1099                  "business_impact_documented"
                  ],
1100                "params_schema": {
1101                  "required": [
                      "ticket_id",
1102                    "escalation_reason",
1103                    "desired_resolution_timeframe"
                    ]
1104                }
1105              },
              {
1106                "id": "update_internal_guidance_document",
1107                "type": "update_document",
                  "description": "Revise internal guidance materials to reflect recent procedural changes.",
1108                "constraints": [
1109                  "approval_for_edits",
                    "current_policy_reviewed"
1110                ],
1111                "params_schema": {
                    "required": [
1112                    "document_id",
1113                    "sections_to_update",
                      "revision_notes"
1114                  ]
1115                }
              },
1116              {
1117                "id": "share_internal_best_practices_document",
                  "type": "share_document",
1118                "description": "Distribute a document outlining internal best practices to the wider department
1119            .",
                  "constraints": [
1120                  "document_reviewed",
1121                  "distribution_list_confirmed"
                  ],
1122                "params_schema": {
1123                  "required": [
                      "document_id",
1124                    "recipients",
1125                    "share_message"
                    ]
1126                }
1127              },
              {
1128                "id": "update_project_milestone_tracker",
1129                "type": "update_project_plan",
                  "description": "Revise the project milestone tracker to reflect recent changes in timelines.",
1130                "constraints": [
1131                  "latest_project_updates_collected",
                    "tracker_accessible"
1132                ],
                  "params_schema": {
1133                  "required": [
                      "project_id",
                      "milestone_updates",
                      "responsible_owners"
                    ]
                  }
              },
              {
                "id": "request_access_to_new_analytics_tools",
                "type": "request_access",
                "description": "Submit an access request for recently deployed analytics platforms to support
            project work.",
```

```
1134          "constraints": [
1135            "manager_approval_obtained",
1136            "tool_list_reviewed"
              ],
1137          "params_schema": {
1138            "required": [
1139              "tool_name",
                 "user_id",
1140              "business_justification"
              ]
1141          }
            },
1142        {
            "id": "draft_feedback_on_proposed_project_approach",
1143        "type": "provide_feedback",
            "description": "Provide written feedback on the proposed approach for a key project based on
1144         available guidance.",
1145        "constraints": [
              "guidance_document_accessible",
1146          "recent_project_overview_reviewed"
            ],
1147        "params_schema": {
1148          "required": [
              "feedback_recipient",
1149          "project_id",
              "feedback_content"
1150          ]
          }
1151      }
        ],
1152    "organizational_structure": {
          "company_name": "PwC UK",
1153      "department": "Financial Services Lead Advisory (FSLA)",
1154      "team_size": 16,
          "reporting_to": "Director, Financial Services Lead Advisory",
1155      "key_meetings": [
1156        "Weekly FSLA team standup",
            "Monday cross-sector M&A pipeline review",
1157        "Bi-weekly Portfolio Advisory Group sync",
1158        "Monthly Financial Services Risk Committee",
            "Quarterly Executive Steering Committee",
1159        "Client transaction strategy sessions",
            "Regulatory and Compliance update",
1160        "Global FSLA practice call",
            "Cross-regional deal origination forum",
1161        "Ad hoc project war room"
          ]
1162    },
        "context_difficulty": "hard"
1163    }
      }
1164
```

Listing 1: Complete world model example for Connor Smith

### Bottleneck Examples

The following JSON structure represents a complete bottleneck from our benchmark dataset.
Bottlenecks are often multifasceted and require information stemming from multiple sources.

```
"bottleneck": {
  "description": "A critical data extract from Sophie Laurent's team at DataInsight Analytics,
    required by the risk-adjusted NPL valuation report (needed during Monday's IC review), was
    delivered in an incompatible file format (SAS instead of Excel), and your IT support ticket (#PWCUK
    -90812) to convert the file has been open 3 days without response.",
  "evidence_required": [
      "document_fb96fa16",
      "document_46b3383d",
      "document_236c752a",
      "document_6ea213cf",
      "document_982d62dc"
      ],
}
```

Listing 2: Complete bottleneck example for Connor Smith

## C APPENDIX C: DATA GENERATION PROMPTS

This appendix contains all prompt templates used in the PROBE evaluation pipeline for generating
synthetic evaluation data. The pipeline consists of five main stages, each with its own set of prompts:

1. **World Model Generation**: Creates comprehensive context from LinkedIn personas

2. **Bottleneck Injection**: Generates realistic productivity bottlenecks

3. **Checklist Generation**: Creates three-step evaluation checklists

4. **True Positive Generation**: Generates corpus items containing evidence

5. **Distractor Generation**: Creates plausible but irrelevant corpus items

Each prompt is designed as a Jinja2 template, allowing dynamic content insertion based on the evaluation context.

## C.1 WORLD MODEL GENERATION PROMPTS

The World Model Generator creates comprehensive professional contexts from LinkedIn personas, including relationships, personal context, available actions, and organizational structure.

### C.1.1 GENERATE ACTIONS FOR BOTTLENECK

**Usage Context**

This prompt is used to generate proactive actions that a persona can take to address specific bottlenecks. It runs after bottlenecks have been identified and creates action items that are contextually appropriate for the persona's role and organization.

```
You are tasked with generating proactive actions for a professional based on specific bottlenecks they
    face.

PERSONA INFORMATION:
- Name: {{ persona.name }}
- Occupation: {{ persona.occupation }}
- Location: {{ persona.location }}
- About: {{ persona.about }}

ORGANIZATION CONTEXT:
- Company: {{ org_structure.company_name }}
- Department: {{ org_structure.department }}
- Team Size: {{ org_structure.team_size }}
- Reports To: {{ org_structure.reporting_to }}

BOTTLENECKS TO ADDRESS:
{% for bottleneck in bottlenecks %}
{{ loop.index }}. {{ bottleneck.description }}
{% endfor %}

DIFFICULTY LEVEL: {{ difficulty }}

## CRITICAL REQUIREMENTS:
1. Generate exactly {{ num_actions }} proactive actions total
2. **EXACTLY ONE ACTION** should solve each bottleneck - no more, no less
3. The remaining actions should be realistic and detailed workplace actions that DON'T solve any of the
    bottlenecks
4. Make it clear which action solves which bottleneck through the action's description and parameters
5. **NO NAMED ENTITIES**: Actions must NOT contain specific person names, company names, or proper nouns
    from the bottlenecks

## ACTION CATEGORIES:
- send_email: Send new emails to individuals or groups
- reply_email: Reply to existing email threads
- schedule_meeting: Create new meetings or events
- reschedule_meeting: Move or modify existing meetings
- cancel_meeting: Cancel scheduled meetings
- create_task: Create new tasks or tickets
- delegate_task: Assign tasks to team members
- update_task_status: Update progress on existing tasks
- create_document: Create new documents, reports, or presentations
- update_document: Edit or revise existing documents
- share_document: Share documents with stakeholders
- send_slack_message: Send instant messages via Slack
- make_phone_call: Initiate phone calls
- request_access: Request access to systems or resources
- provide_feedback: Give feedback on work or proposals
- request_approval: Ask for sign-offs or approvals
- escalate_to_manager: Escalate issues up the chain
- create_meeting_agenda: Prepare agenda for meetings
- conduct_review: Perform code or document reviews
- update_project_plan: Modify project timelines or scope
```

```
1242
1243    OUTPUT FORMAT:
        Return a JSON object with an "actions" array. Each action should follow this structure:
1244
        {
1245      "actions": [
            {
1246          "id": "unique_action_identifier",
              "type": "action_category",
1247          "description": "Clear description of what this action does",
              "constraints": ["Array of preconditions or policies this action must respect"],
1248          "params_schema": {
                "required": ["Array of required parameter names"]
1249          },
              "solves_bottleneck": null or bottleneck_index (1-based index if this action solves a bottleneck)
1250        }
          ]
1251    }
1252
        {% if difficulty == "easy" %}
1253    For EASY difficulty:
        - Actions that solve bottlenecks should be straightforward and obvious
1254    - Include simple parameters like "recipient", "subject", "content"
        - Non-bottleneck actions should be basic routine tasks
1255    {% elif difficulty == "medium" %}
        For MEDIUM difficulty:
1256    - Actions that solve bottlenecks should require some thought, ids should avoid using bottleneck keywords
              .
1257    - Include parameters like "priority", "stakeholders", "deadline", "approach"
        - Non-bottleneck actions should be moderately complex coordination tasks
1258    - All action descriptions should be a bit general, and not mention the bottleneck or its details in any
              way
1259    {% elif difficulty == "hard" %}
        For HARD difficulty:
1260    - Actions that solve bottlenecks should be subtle and ids should avoid using bottleneck keywords.
        - Non-bottleneck actions should be strategic and cross-functional
1261    - All action descriptions should be somewhat general and vague, and not mention the bottleneck or its
              details in any way
1262    {% endif %}
1263
        EXAMPLE for a bottleneck about "Email to David Kim about security audit findings remains unanswered, and
1264          he does not have the authority to approve the security audit findings":
        {
1265      "actions": [
            {
1266          "id": "schedule_team_meeting",
              "type": "schedule_meeting",
1267          "description": "Schedule a regular team meeting to discuss project updates and coordination.",
              "constraints": ["team_availability", "meeting_room_available"],
1268          "params_schema": {
                "required": ["attendees", "date", "time", "agenda", "location"]
1269          },
              "solves_bottleneck": null
1270        },
            {
1271          "id": "update_project_status",
              "type": "update_document",
1272          "description": "Update project status documentation with current progress and milestones.",
              "constraints": ["document_access", "accurate_information"],
1273          "params_schema": {
                "required": ["document_id", "status_update", "completion_percentage", "next_steps"]
1274          },
              "solves_bottleneck": null
1275        },
            {
1276          "id": "send_weekly_report",
              "type": "send_email",
1277          "description": "Send weekly progress report to stakeholders and team members.",
              "constraints": ["report_data_available", "stakeholder_list_current"],
1278          "params_schema": {
                "required": ["recipients", "subject", "report_content", "attachments"]
1279          },
              "solves_bottleneck": null
1280        },
            {
1281          "id": "conduct_code_review",
              "type": "conduct_review",
1282          "description": "Review code changes submitted by team members for quality and standards compliance
              .",
1283          "constraints": ["technical_expertise", "time_available"],
              "params_schema": {
1284            "required": ["pull_request_id", "review_criteria", "feedback_type", "approval_status"]
              },
1285          "solves_bottleneck": null
            },
1286          {
              "id": "escalate_issue",
1287          "type": "escalate_to_manager",
              "description": "Escalate an important issue to management for resolution.",
1288          "constraints": ["multiple_attempts_made", "deadline_approaching", "requires_higher_authority"],
              "params_schema": {
1289            "required": ["original_recipient", "escalation_recipient", "urgency_level", "business_impact", "
              attempted_contacts"]
```

```
      },
      "solves_bottleneck": 1
    },
    {
      "id": "delegate_routine_task",
      "type": "delegate_task",
      "description": "Delegate routine tasks to appropriate team members to optimize workload
      distribution.",
      "constraints": ["team_capacity", "skill_match"],
      "params_schema": {
        "required": ["assignee", "task_description", "deadline", "priority_level"]
      },
      "solves_bottleneck": null
    }
  ]
}

Ensure that:
1. Each bottleneck has EXACTLY ONE action that can solve it, all other actions should certainly not
      solve the bottleneck
2. The action description of the correct action should address the bottleneck, but without mentioning
      the bottleneck, keywords, or its details in any way.
3. Other actions are detailed and realistic but explicitly DON'T solve any of the listed bottlenecks
4. Total number of actions equals {{ num_actions }}
5. **CRITICAL**: The actions should not include any mention of the people or situations involved in the
      bottleneck
```

Listing 3: generate_actions_for_bottleneck.j2

### C.1.2 GENERATE ORGANIZATION STRUCTURE

**Usage Context**

This prompt generates the organizational context around a persona, including company structure, team composition, reporting lines, and key processes. It's one of the first prompts executed to establish the professional environment.

```
Generate a realistic organizational structure for the following professional:

Name: {{ persona.name }}
Occupation: {{ persona.occupation }}
Location: {{ persona.location }}
About: {{ persona.about }}

Create a detailed organizational context that includes:
1. Company name and type
2. Department structure
3. Team composition
4. Reporting relationships
5. Key processes and workflows

The organization should be realistic for someone in their role and location.

Provide your response as a JSON object with this structure:
{
  "company_name": "Name of the company",
  "company_type": "Type of company (startup, enterprise, etc.)",
  "department": "Their department name",
  "team_size": 5,
  "direct_reports": 2,
  "reporting_to": "Title of their manager",
  "key_processes": ["Process 1", "Process 2"],
  "typical_meetings": ["Meeting type 1", "Meeting type 2"]
}
```

Listing 4: generate_org_structure.j2

### C.1.3 GENERATE PERSONAL CONTEXT

**Usage Context**

This prompt creates personal work context for the persona, including their work style, preferences, current goals, constraints, and tools they use. This adds depth to the persona beyond their LinkedIn profile.

```
Generate personal work context for the following professional:

PERSONA:
- Name: {{ persona.name }}
- Occupation: {{ persona.occupation }}
- About: {{ persona.about }}

DIFFICULTY: {{ difficulty }}

Create realistic personal context including:
1. Work style and preferences
2. Current goals and priorities
3. Time constraints and challenges
4. Communication preferences
5. Tools and systems they use

{% if difficulty == "easy" %}
Keep the context simple with straightforward preferences and minimal constraints.
{% elif difficulty == "medium" %}
Include moderate complexity with some competing priorities and constraints.
{% elif difficulty == "hard" %}
Create complex context with multiple competing priorities, significant constraints, and nuanced
    preferences.
{% endif %}

Provide your response as a JSON object with this structure:
{
  "work_style": "Description of how they prefer to work",
  "current_goals": ["Goal 1", "Goal 2", "Goal 3"],
  "constraints": ["Time constraint", "Resource constraint"],
  "communication_preferences": "How they prefer to communicate",
  "tools_used": ["Tool 1", "Tool 2"],
  "peak_productivity_time": "When they work best",
  "biggest_challenges": ["Challenge 1", "Challenge 2"]
}
```

Listing 5: generate_personal_context.j2

### C.1.4  GENERATE RELATIONSHIPS

> **Usage Context**
>
> This prompt generates professional relationships for the persona, including colleagues, clients, stakeholders, and collaborators. The number and complexity of relationships varies based on the difficulty level.

```
Generate professional relationships for the following person:

Name: {{ persona.name }}
Occupation: {{ persona.occupation }}
Location: {{ persona.location }}
About: {{ persona.about }}

DIFFICULTY LEVEL: {{ difficulty }}

{% if difficulty == "easy" %}
Generate 3-5 key relationships that are straightforward and clearly defined.
{% elif difficulty == "medium" %}
Generate 5-8 relationships with moderate complexity and some overlapping responsibilities.
{% elif difficulty == "hard" %}
Generate 8-12 relationships with complex interdependencies and nuanced dynamics.
{% endif %}

For each relationship, provide:
1. The person's full name
2. Their role/title
3. Type of relationship (colleague, client, manager, stakeholder, collaborator)
4. How they interact with {{ persona.name }}
5. Current status of the relationship

Provide your response as a JSON object with this structure:
{
  "relationships": [
    {
      "name": "Full name",
      "role": "Their job title",
      "type": "colleague|client|manager|stakeholder|collaborator",
      "interaction": "Description of how they work together",
      "status": "Current state of the relationship",
      "frequency": "How often they interact"
    }
  ]
}
```

---

Listing 6: generate_relationships.j2

## C.2   BOTTLENECK INJECTION PROMPTS

The Bottleneck Injector creates realistic productivity bottlenecks that can be addressed by the persona's available actions.

### C.2.1   GENERATE INDIVIDUAL BOTTLENECK

> **Usage Context**
>
> This prompt generates a single, highly specific bottleneck for a persona. It's called multiple times to create a set of bottlenecks, each focusing on a different aspect of the persona's work challenges.

```
You are creating a SINGLE, highly specific productivity bottleneck for {{ persona_name }}.

CONTEXT:
- Occupation: {{ persona_occupation }}
- About: {{ persona_about }}
- Difficulty: {{ difficulty }}

ORGANIZATION:
- Company: {{ org_structure.company_name }}
- Department: {{ org_structure.department }}
- Team Size: {{ org_structure.team_size }}
- Reports To: {{ org_structure.reporting_to }}

KEY RELATIONSHIPS:
{% for rel in relationships[:5] %}
- {{ rel.name }} ({{ rel.role }}): {{ rel.interaction }}
{% endfor %}

PERSONAL CONTEXT:
- Work Style: {{ personal_context.work_style }}
- Current Goals: {{ personal_context.current_goals[:3] | join(', ') }}
- Constraints: {{ personal_context.constraints | join(', ') }}

BOTTLENECK #{{ bottleneck_index }}

Create ONE specific bottleneck that:
1. References REAL NAMES from the relationships
2. Mentions SPECIFIC documents, meetings, or deadlines
3. Has a clear timeline or urgency
4. Can be discovered through search/investigation
5. Is solvable through proactive action

{% if difficulty == "easy" %}
Make it straightforward with clear cause and solution.
{% elif difficulty == "medium" %}
Include some complexity and multiple stakeholders.
{% elif difficulty == "hard" %}
Make it complex with competing priorities and hidden dependencies.
{% endif %}

The bottleneck should be 2-3 sentences maximum and extremely specific.

Example format:
"The Q3 product roadmap review with Sarah Chen is scheduled for next Tuesday, but the feature
    prioritization matrix she requested hasn't been updated since July because the engineering
    estimates from Michael Park's team are still pending in JIRA tickets ENG-4521 through ENG-4525."

Generate a single bottleneck description:
```

Listing 7: generate_bottleneck.j2

### C.2.2 GENERATE BOTTLENECKS BATCH

> **Usage Context**
>
> This prompt generates multiple bottlenecks in a single LLM call for efficiency. It ensures variety across different work aspects while maintaining consistency with the persona's context.

```
You are creating {{ num_bottlenecks }} highly specific productivity bottlenecks for {{ persona_name }}.

CONTEXT:
- Occupation: {{ persona_occupation }}
- About: {{ persona_about }}
- Difficulty: {{ difficulty }}

ORGANIZATION:
- Company: {{ org_structure.company_name }}
- Department: {{ org_structure.department }}
- Team Size: {{ org_structure.team_size }}

KEY RELATIONSHIPS:
{% for rel in relationships %}
- {{ rel.name }} ({{ rel.role }}): {{ rel.interaction }}
{% endfor %}

PERSONAL CONTEXT:
- Work Style: {{ personal_context.work_style }}
- Current Goals: {{ personal_context.current_goals | join(', ') }}
- Constraints: {{ personal_context.constraints | join(', ') }}

Generate {{ num_bottlenecks }} DIFFERENT bottlenecks that:
1. Each references REAL NAMES from the relationships
2. Mentions SPECIFIC artifacts (documents, meetings, systems)
3. Has clear urgency or timeline
4. Can be discovered through search
5. Is solvable through action

ENSURE VARIETY:
- Different types of problems (delays, missing info, conflicts, etc.)
- Different people involved
- Different urgency levels
- Different solutions needed

{% if difficulty == "easy" %}
Make them straightforward with clear causes.
{% elif difficulty == "medium" %}
Include moderate complexity.
{% elif difficulty == "hard" %}
Make them complex with hidden dependencies.
{% endif %}

Provide your response as a JSON object:
{
  "bottlenecks": [
    {
      "description": "Specific 2-3 sentence bottleneck",
      "primary_person": "Main person involved",
      "urgency": "high|medium|low",
      "type": "delay|missing_info|conflict|approval|resource|coordination"
    }
  ]
}
```

Listing 8: generate_bottlenecks_batch.j2

### C.3 CHECKLIST GENERATION PROMPTS

The Checklist Generator creates three-step evaluation checklists that test an agent's ability to complete the proactive workflow.

### C.3.1 THREE-STEP CHECKLIST

> **Usage Context**
>
> This prompt generates the core three-step checklist (Search âĘŠ Identification âĘŠ Task Selection) for evaluating agent performance on a specific bottleneck. We note that these steps need not happen concurrently: we only perform evaluation on the final output.

```
Generate a three-step checklist for addressing the following bottleneck:

BOTTLENECK:
{{ bottleneck.description }}

WORLD MODEL CONTEXT:
- Persona: {{ world_model.persona_full_name }} ({{ world_model.persona_occupation }})
- Company: {{ world_model.organizational_structure.company_name }}
- Difficulty: {{ difficulty }}

KEY RELATIONSHIPS:
{% for rel in world_model.relationships[:5] %}
- {{ rel.name }} ({{ rel.role }}): {{ rel.interaction }}
{% endfor %}

AVAILABLE ACTIONS:
{% for action in available_actions[:8] %}
- {{ action.action_type }}: {{ action.name }}
{% endfor %}

Create a three-step checklist with:

STEP 1 - SEARCH: What specific information should be searched for?
- Include 3-5 specific search queries or data sources
- Reference actual names, documents, or systems from the bottleneck
- Mix of different search types (emails, documents, calendar, etc.)

STEP 2 - IDENTIFICATION: What key insights should be identified?
- 2-3 specific findings that reveal the root cause
- Reference actual evidence that would be found
- Clear connection to the bottleneck

STEP 3 - TASK SELECTION: What action should be taken?
- Select from available actions
- Include specific parameters (who, what, when)
- Clear resolution to the bottleneck

Provide your response as a JSON object:
{
  "search_step": {
    "description": "What to search for and why",
    "specific_queries": [
      "Query 1 with actual names/docs",
      "Query 2 with specific terms",
      "Query 3 with system references"
    ],
    "expected_sources": ["email", "calendar", "documents"]
  },
  "identification_step": {
    "description": "What insights to identify",
    "key_findings": [
      "Specific finding 1",
      "Specific finding 2"
    ],
    "root_cause": "The underlying issue"
  },
  "task_selection_step": {
    "action_type": "One of the available action types",
    "description": "Specific action to take",
    "parameters": {
      "participants": ["Names"],
      "timeline": "When",
      "deliverables": "What"
    }
  }
}
```

Listing 9: three_step_checklist.j2

### C.4 TRUE POSITIVE GENERATION PROMPTS

These prompts generate corpus items that contain evidence of bottlenecks, serving as the "ground truth" that agents should find.

### C.4.1 PLAN EVIDENCE DISTRIBUTION

> **Usage Context**
>
> This prompt plans how evidence for a bottleneck will be distributed across multiple corpus items. It ensures comprehensive coverage while avoiding contamination from other bottlenecks.

```
You are planning how to distribute evidence for a bottleneck across multiple documents.

BOTTLENECK TO ADDRESS:
{{ bottleneck.description }}

WORLD MODEL CONTEXT:
- Persona: {{ world_model.persona_full_name }} ({{ world_model.persona_occupation }})
- Company: {{ world_model.organizational_structure.company_name }}

KEY RELATIONSHIPS:
{% for rel in world_model.relationships[:5] %}
- {{ rel.name }} ({{ rel.role }})
{% endfor %}

AVAILABLE DOCUMENT TYPES:
- Email (conversations, requests, updates)
- Calendar (meetings, deadlines, events)
- Document (reports, plans, specifications)

OTHER BOTTLENECKS TO AVOID:
{% for other in other_bottlenecks %}
- {{ other }}
{% endfor %}

Plan how to distribute evidence across {{ num_documents }} documents:
1. Each document should contain a different aspect/angle of the bottleneck
2. Together they should tell the complete story
3. Avoid ANY mention of other bottlenecks
4. Make evidence discoverable but not too obvious

Provide your response as a JSON object:
{
  "evidence_distribution": [
    {
      "document_type": "email|calendar|document",
      "evidence_role": "What aspect this covers",
      "key_information": "Specific info to include",
      "sender_or_creator": "Who creates this",
      "discoverability": "How someone would find this"
    }
  ]
}
```

Listing 10: plan_evidence_distribution.j2

### C.4.2 GENERATE EMAIL EVIDENCE

> **Usage Context**
>
> This prompt generates email corpus items that contain evidence of the bottleneck. Emails often contain requests, updates, and clarifications that reveal bottleneck details.

```
Generate a realistic email that contains evidence of the following bottleneck:

BOTTLENECK:
{{ bottleneck.description }}

EVIDENCE ROLE:
This email should specifically show: {{ evidence_role }}

WORLD MODEL:
- Persona: {{ world_model.persona_full_name }} ({{ world_model.persona_occupation }})
- Email: {{ world_model.persona_full_name.lower().replace(' ', '.') }}@{{ world_model.
    organizational_structure.company_name.lower().replace(' ', '').replace(',', '') }}.com

KEY RELATIONSHIPS:
{% for rel in world_model.relationships[:7] %}
- {{ rel.name }} ({{ rel.role }})
{% endfor %}
```

```
THINGS TO AVOID:
Do NOT mention or reference these other bottlenecks:
{% for other in other_bottlenecks %}
{{ other }}
{% endfor %}

Generate a complete, realistic email that:
1. Contains clear evidence of the bottleneck
2. Fits the evidence role specified
3. Includes realistic email metadata
4. Uses actual names from relationships
5. Avoids any mention of other bottlenecks
6. Sounds natural and professional

The email should be substantial (200-400 words) and include:
- Proper email headers (From, To, CC, Subject, Date)
- Natural greeting and sign-off
- Specific details that reveal bottleneck information
- Realistic workplace communication style

Format as:
From: sender@company.com
To: recipient@company.com
CC: others@company.com
Subject: Specific subject line
Date: Recent date

Email body...
```

Listing 11: generate_email_evidence.j2

### C.4.3 GENERATE CALENDAR EVIDENCE

> **Usage Context**
>
> This prompt generates calendar events that reveal scheduling conflicts, deadlines, or meeting-related bottleneck evidence.

```
Generate a realistic calendar event that contains evidence of the following bottleneck:

BOTTLENECK:
{{ bottleneck.description }}

EVIDENCE ROLE:
This calendar event should show: {{ evidence_role }}

WORLD MODEL:
- Persona: {{ world_model.persona_full_name }}
- Company: {{ world_model.organizational_structure.company_name }}

KEY PEOPLE:
{% for rel in world_model.relationships[:5] %}
- {{ rel.name }} ({{ rel.role }})
{% endfor %}

AVOID MENTIONING:
{% for other in other_bottlenecks %}
- {{ other }}
{% endfor %}

Create a detailed calendar event that:
1. Reveals important timing/scheduling aspects of the bottleneck
2. Includes realistic attendees from relationships
3. Has detailed agenda or description
4. Shows urgency or conflicts if relevant
5. Completely avoids other bottlenecks

Include:
- Title: Specific and professional
- Date/Time: Realistic and relevant to bottleneck
- Duration: Appropriate for the meeting type
- Location: Physical or virtual
- Attendees: Mix of required and optional
- Agenda/Description: Detailed and revealing bottleneck evidence
- Any attached documents or pre-reads

Format your response as a complete calendar event.
```

Listing 12: generate_calendar_evidence.j2

### C.4.4 GENERATE DOCUMENT EVIDENCE

> **Usage Context**
>
> This prompt generates longer documents (reports, plans, memos) that contain comprehensive evidence about the bottleneck, often providing context and history.

```
Generate a professional document that contains evidence of the following bottleneck:

BOTTLENECK:
{{ bottleneck.description }}

EVIDENCE ROLE:
This document should provide: {{ evidence_role }}

CONTEXT:
- Author: {{ world_model.persona_full_name }}
- Organization: {{ world_model.organizational_structure.company_name }}
- Department: {{ world_model.organizational_structure.department }}

DOCUMENT REQUIREMENTS:
- Length: {{ min_words }}-{{ max_words }} words
- Type: Report, memo, plan, or specification
- Should reveal key bottleneck information
- Must seem like a natural workplace document

KEY PEOPLE TO REFERENCE:
{% for rel in world_model.relationships[:6] %}
- {{ rel.name }} ({{ rel.role }})
{% endfor %}

STRICTLY AVOID:
{% for other in other_bottlenecks %}
- {{ other }}
{% endfor %}

Generate a complete professional document that:
1. Has proper header (title, date, author, recipients)
2. Contains multiple sections with clear headings
3. Embeds bottleneck evidence naturally throughout
4. References real people and specific details
5. Maintains professional tone and formatting
6. Includes actionable information
7. Never mentions other bottlenecks

The document should read like an authentic workplace artifact that someone would search for when
    investigating the bottleneck.
```

Listing 13: generate_document_evidence.j2

### C.4.5 GENERATE DYNAMIC SOURCES

> **Usage Context**
>
> This prompt identifies additional data sources or systems where evidence might be found, expanding beyond the standard email/calendar/document trinity.

```
Identify specific data sources where evidence for this bottleneck would be found:

BOTTLENECK:
{{ bottleneck.description }}

ORGANIZATION:
- Company: {{ world_model.organizational_structure.company_name }}
- Industry: {{ world_model.organizational_structure.company_type }}
- Department: {{ world_model.organizational_structure.department }}

Suggest 3-5 specific systems, databases, or specialized sources where evidence would exist.

For each source:
1. Name the specific system/platform
2. What evidence would be found there
3. How to search/access it
4. Why it's relevant to this bottleneck

Examples: JIRA tickets, Confluence pages, Slack channels, CRM records, Github PRs, etc.

Provide specific names and identifiers, not generic categories.
```

Listing 14: generate_dynamic_sources.j2

### C.4.6 REVIEW EVIDENCE

> **Usage Context**
>
> This prompt is used to review generated evidence for quality, ensuring it properly supports the bottleneck discovery without contamination from other bottlenecks.

```
Review the following evidence for quality and effectiveness:

BOTTLENECK BEING EVIDENCED:
{{ bottleneck.description }}

GENERATED EVIDENCE:
{{ evidence_content }}

OTHER BOTTLENECKS TO AVOID:
{% for other in other_bottlenecks %}
- {{ other }}
{% endfor %}

Evaluate:
1. Does the evidence clearly support discovering this bottleneck?
2. Is it discoverable through realistic search queries?
3. Does it avoid ALL mentions of other bottlenecks?
4. Is it natural and realistic for the workplace context?
5. Are all names, dates, and details consistent?

Provide:
1. Quality score (1-10)
2. Strengths of the evidence
3. Any issues or contamination found
4. Suggested improvements
5. Search queries that would find this evidence

Format as JSON:
{
  "quality_score": 8,
  "strengths": ["Clear timeline", "Specific names"],
  "issues": ["Might be too obvious"],
  "improvements": ["Add more context about..."],
  "search_queries": ["Michael Park ENG-4521", "Q3 roadmap review"]
}
```

Listing 15: review_evidence.j2

### C.5 DISTRACTOR GENERATION PROMPTS

These prompts generate plausible but irrelevant corpus items that test the agent's ability to filter out noise.

### C.5.1 GENERATE EMAIL DISTRACTORS

> **Usage Context**
>
> This prompt generates realistic workplace emails that are plausible distractors - they should seem relevant to the persona's work but not contain evidence of any bottlenecks.

```
Generate {{ count }} realistic workplace emails for {{ world_model.persona_full_name }}'s context.

CONTEXT:
- Role: {{ world_model.persona_occupation }}
- Company: {{ world_model.organizational_structure.company_name }}
- Department: {{ world_model.organizational_structure.department }}

RELATIONSHIPS TO USE:
{% for rel in world_model.relationships %}
- {{ rel.name }} ({{ rel.role }}): {{ rel.interaction }}
{% endfor %}
```

```
CRITICAL - AVOID ALL BOTTLENECKS:
{% for bottleneck in bottlenecks %}
 {{ bottleneck.description }}
{% endfor %}

REQUIREMENTS:
1. Emails must be completely unrelated to any bottleneck
2. Should be realistic workplace communications
3. Vary the types: updates, requests, FYIs, discussions
4. Use different senders and recipients
5. Include realistic dates and subjects
6. Length: 150-300 words each

Generate diverse emails about:
- Routine status updates
- General team communications
- Company announcements
- Non-critical planning
- Social/cultural events
- Training or development
- General process discussions

Ensure NONE of the emails could be interpreted as evidence for any bottleneck.

Format each email with proper headers (From, To, Subject, Date) followed by the body.
```

Listing 16: generate_email_distractors.j2

### C.5.2 GENERATE CALENDAR DISTRACTORS

> **Usage Context**
>
> This prompt creates calendar events that represent normal workplace meetings and events,
> serving as noise that agents must filter through.

```
Generate {{ count }} realistic calendar events for {{ world_model.persona_full_name }}.

CONTEXT:
- Role: {{ world_model.persona_occupation }}
- Company: {{ world_model.organizational_structure.company_name }}
- Typical Meetings: {{ world_model.organizational_structure.typical_meetings | join(', ') }}

PEOPLE TO INCLUDE:
{% for rel in world_model.relationships[:8] %}
- {{ rel.name }} ({{ rel.role }})
{% endfor %}

MUST AVOID - NO BOTTLENECK EVIDENCE:
{% for bottleneck in bottlenecks %}
 {{ bottleneck.description }}
{% endfor %}

Create diverse calendar events:
1. Regular recurring meetings (1-on-1s, team standups)
2. Training or development sessions
3. Company-wide events
4. Social activities
5. Planning sessions (unrelated to bottlenecks)
6. Reviews or retrospectives

Each event needs:
- Title: Professional and specific
- Date/Time: Spread across different days/times
- Duration: Realistic for the meeting type
- Attendees: Appropriate mix of people
- Location/Link: Physical or virtual
- Description: Detailed agenda that contains NO bottleneck evidence

Make them indistinguishable from real important meetings but completely unrelated to bottlenecks.
```

Listing 17: generate_calendar_distractors.j2

### C.5.3  GENERATE DOCUMENT DISTRACTORS

**Usage Context**

This prompt generates longer-form documents that serve as distractors, representing typical workplace documentation that doesn't relate to any bottlenecks.

```
Generate {{ count }} professional documents for {{ world_model.persona_full_name }}'s work context.

CONTEXT:
- Role: {{ world_model.persona_occupation }}
- Department: {{ world_model.organizational_structure.department }}
- Company: {{ world_model.organizational_structure.company_name }}

DOCUMENT TYPES TO CREATE:
1. Process documentation
2. Team updates or newsletters
3. Project proposals (unrelated to bottlenecks)
4. Meeting notes
5. Training materials
6. Policy documents

CRITICAL - AVOID ALL BOTTLENECKS:
{% for bottleneck in bottlenecks %}
 DO NOT REFERENCE: {{ bottleneck.description }}
{% endfor %}

Each document should:
- Be 400-800 words
- Have professional formatting and structure
- Reference real people from relationships
- Contain valuable but irrelevant information
- Be discoverable by plausible search terms
- Seem important enough to not ignore

Include proper headers:
- Title
- Author
- Date
- Document type
- Recipients/Audience

The documents should be high-quality distractors that would naturally appear in search results but
    provide no evidence for any bottleneck.
```

Listing 18: generate_document_distractors.j2

### C.5.4  GENERATE NATURAL DISTRACTOR

**Usage Context**

This is a general-purpose prompt for generating natural distractors of any type, with emphasis on making them realistic and contextually appropriate.

```
Generate a natural {{ kind }} distractor for the following context:

PERSONA: {{ world_model.persona_full_name }} ({{ world_model.persona_occupation }})
COMPANY: {{ world_model.organizational_structure.company_name }}

AVAILABLE RELATIONSHIPS:
{% for rel in world_model.relationships[:6] %}
- {{ rel.name }} ({{ rel.role }})
{% endfor %}

WORK CONTEXT:
- Department: {{ world_model.organizational_structure.department }}
- Current Goals: {{ world_model.personal_context.current_goals[:3] | join(', ') }}
- Tools Used: {{ world_model.personal_context.tools_used | join(', ') }}

MUST AVOID (NO EVIDENCE OF):
{% for bottleneck in bottlenecks %}
- {{ bottleneck.description }}
{% endfor %}

Create a {{ kind }} that:
1. Is completely unrelated to any bottleneck
2. Fits naturally in the persona's work life
3. Could plausibly be important
```

```
4. Uses real names and realistic details
5. Matches typical {{ kind }} format and style

Focus on routine work activities that would generate {{ kind }}s but don't relate to the specific
    problems being evaluated.
```

Listing 19: generate_natural_distractor.j2

### C.5.5    ENHANCE DISTRACTOR

**Usage Context**

This prompt enhances basic distractors to make them more realistic and harder to distinguish from true positives, adding details and context.

```
Enhance the following distractor to make it more realistic and detailed:

ORIGINAL DISTRACTOR:
{{ original_content }}

CONTEXT:
- Type: {{ distractor_type }}
- Persona: {{ world_model.persona_full_name }}
- Company: {{ world_model.organizational_structure.company_name }}

ENHANCEMENT GOALS:
1. Add more specific details (names, dates, numbers)
2. Include realistic workplace jargon
3. Add urgency or importance markers
4. Reference real systems or processes
5. Make it harder to distinguish from true positives

MAINTAIN:
- Core message/purpose
- Avoidance of all bottlenecks
- Professional tone
- Realistic length

The enhanced version should be a high-quality distractor that requires careful analysis to determine it'
    s not relevant to the bottlenecks.
```

Listing 20: enhance_distractor.j2

### C.6    LLM-AS-JUDGE EVALUATION PROMPTS

**Usage Context**

This prompt is used to for evaluating bottleneck identification as described in section 3

```
You are evaluating whether an AI agent correctly identified a bottleneck using precision-based scoring.

## Context
The agent was given various documents and data sources to analyze, and needed to identify the bottleneck
    affecting productivity.

## Workplace Context
Persona: {{ world_model.persona_full_name }}
Company: {{ world_model.organizational_structure.company_name }}
Department: {{ world_model.organizational_structure.department }}
Team Size: {{ world_model.organizational_structure.team_size }}

## What the Agent Identified
"{{ identified_bottleneck }}"

## The Actual Bottleneck
"{{ actual_bottleneck }}"

## Two-Tier Precision Scoring System

### ESSENTIAL DETAILS (Must be 100% correct for any points)
These are execution-critical details required for immediate action:

1. **WHO_BLOCKED**: The exact person/entity being blocked (specific names, not roles)
2. **WHO_BLOCKER**: The exact person/entity causing the blockage (specific names, not roles)
3. **WHAT_TASK**: The specific deliverable/work item that's blocked (exact names, not generic
    descriptions)
```

```
4. **WHY_ROOT_CAUSE**: The true underlying reason for the blockage (not symptoms)

### NON-ESSENTIAL DETAILS (Can have minor errors/omissions)
These provide context but aren't critical for immediate resolution:

5. **WHEN_DEADLINE**: Specific dates and timelines
6. **WHERE_SYSTEM**: Exact system/tool names and locations
7. **HOW_MECHANISM**: Detailed process or workflow information
8. **IMPACT_SCOPE**: Downstream effects and dependencies

## Scoring Rules

**CORRECT (1.0)**: All essential details are accurate AND all non-essential details are accurate
**PARTIALLY_CORRECT (0.5)**: All essential details are accurate BUT has errors/omissions in non-
     essential details
**INCORRECT (0.0)**: Any essential detail is wrong, missing, or too vague

## Essential Detail Requirements
- **Names must be specific**: "Timothy" not "someone from finance"
- **Systems must be exact**: "econo.com" not "financial system"
- **Tasks must be precise**: "Q3 financials initiative" not "quarterly report"
- **Root causes must be accurate**: "Martha stripped access" not "system issues"

## Examples

### Example 1: CORRECT (1.0)
**Actual**: Timothy isn't meeting the March 23rd deadline for his Q3 financials initiative because
     Martha revoked his econo.com access to company financials.

**Agent**: Timothy cannot complete his Q3 financials initiative by the March 23rd deadline because
     Martha removed his access to company financials through econo.com.

**Analysis**:
- âIJĖ WHO_BLOCKED: Timothy (correct)
- âIJĖ WHO_BLOCKER: Martha (correct)
- âIJĖ WHAT_TASK: Q3 financials initiative (correct)
- âIJĖ WHY_ROOT_CAUSE: Martha revoked access (correct)
- âIJĖ All non-essential details accurate

### Example 2: PARTIALLY_CORRECT (0.5)
**Actual**: Timothy isn't meeting the March 23rd deadline for his Q3 financials initiative because
     Martha revoked his econo.com access to company financials.

**Agent**: Timothy cannot complete his Q3 financials initiative because Martha removed his access to
     company financial systems.

**Analysis**:
- âIJĖ WHO_BLOCKED: Timothy (correct)
- âIJĖ WHO_BLOCKER: Martha (correct)
- âIJĖ WHAT_TASK: Q3 financials initiative (correct)
- âIJĖ WHY_ROOT_CAUSE: Martha revoked access (correct)
- âÎŇ WHEN_DEADLINE: Missing March 23rd
- âÎŇ WHERE_SYSTEM: "financial systems" instead of "econo.com"

### Example 3: INCORRECT (0.0)
**Actual**: Timothy isn't meeting the March 23rd deadline for his Q3 financials initiative because
     Martha revoked his econo.com access to company financials.

**Agent**: Someone from finance is having trouble with their quarterly report due to system access
     issues.

**Analysis**:
- âÎŇ WHO_BLOCKED: "someone from finance" instead of Timothy
- âÎŇ WHO_BLOCKER: Missing Martha entirely
- âÎŇ WHAT_TASK: "quarterly report" instead of Q3 financials initiative
- âÎŇ WHY_ROOT_CAUSE: "system issues" instead of Martha's action

## Output Format
Return a JSON object with:
{
  "judgment": "<CORRECT|PARTIALLY_CORRECT|INCORRECT>",
  "essential_details_analysis": {
    "who_blocked": "<correct|incorrect|missing>",
    "who_blocker": "<correct|incorrect|missing>",
    "what_task": "<correct|incorrect|missing>",
    "why_root_cause": "<correct|incorrect|missing>"
  },
  "non_essential_details_analysis": {
    "when_deadline": "<correct|incorrect|missing|n/a>",
    "where_system": "<correct|incorrect|missing|n/a>",
    "how_mechanism": "<correct|incorrect|missing|n/a>",
    "impact_scope": "<correct|incorrect|missing|n/a>"
  },
  "reasoning": "<explanation of your scoring decision focusing on essential vs non-essential accuracy>"
}
```

Listing 21: judge_bottleneck_identification.j2

**Usage Context**

This prompt is used to for evaluating parameter scoring as part of the task execution evaluation metric as described in section 3

```
You are an expert evaluator specializing in assessing AI agent action parameter selection in workplace
    automation scenarios. Your task is to evaluate whether an AI agent selected appropriate parameters
    for a correctly identified action.

## Your Role and Expertise
You have deep expertise in:
- Workplace automation and task execution
- API parameter design and semantic equivalence
- Business process optimization
- Contextual reasoning in parameter selection

## Evaluation Context

### The Bottleneck Being Resolved
```
{{ bottleneck.description }}
```

### Workplace Environment
- **Company**: {{ world_model.organizational_structure.company_name }}
- **Department**: {{ world_model.organizational_structure.department }}
- **Persona**: {{ world_model.persona_full_name }}

### Key Relationships Available
{% for rel in world_model.relationships[:5] %}
- **{{ rel.name }}** ({{ rel.type.value }}): {{ rel.department if rel.department else "External" }}{% if
    rel.email %} - {{ rel.email }}{% endif %}
{% endfor %}

### Evidence That Led to Bottleneck Discovery
{% if true_positives %}
{% for tp in true_positives[:3] %}
- **{{ tp.get('type', 'Unknown') }}**: {{ tp.get('title', tp.get('subject', 'N/A')) }}
  - Key info: {{ tp.get('summary', tp.get('content', 'Details not shown'))[:100] }}...
{% endfor %}
{% else %}
- No specific evidence items provided for context
{% endif %}

## What to Evaluate

### Selected Action
- **Action Type**: {{ selected_action.action_id }}
- **Purpose**: To resolve the identified bottleneck

### Agent's Selected Parameters
```json
{{ selected_parameters | tojson(indent=2) }}
```

### Expected Parameters (Ground Truth)
```json
{{ expected_parameters | tojson(indent=2) }}
```

## Evaluation Framework

### Step 1: Understand Parameter Intent
For each parameter in the expected set, identify:
1. **Purpose**: What this parameter accomplishes
2. **Criticality**: Is it essential for resolving the bottleneck?
3. **Flexibility**: Can alternatives achieve the same goal?

### Step 2: Map Parameters Semantically
Compare agent's parameters to expected parameters:
1. **Direct matches**: Same parameter name and equivalent value
2. **Semantic matches**: Different representation, same effect
3. **Missing parameters**: Expected but not provided
4. **Extra parameters**: Provided but not expected
5. **Wrong parameters**: Provided but incorrect for the goal

### Step 3: Evaluate Effectiveness
Ask: "Would the agent's parameters successfully resolve the bottleneck?"

## Scoring Rubric

### CORRECT (Score: 1.0)
All of the following must be true:
- âĲŚ All critical parameters are present (directly or semantically)
- âĲŚ Parameter values would achieve the bottleneck resolution
- âĲŚ Any deviations are reasonable improvements or valid alternatives
- âĲŚ No critical information is wrong or missing
- âĲŚ Extra parameters (if any) don't interfere with the goal
```

```
### PARTIALLY_CORRECT (Score: 0.5)
The parameters show understanding but have gaps:
– ⚠️ Most critical parameters present (70-90%)
– ⚠️ Would partially resolve the bottleneck
– ⚠️ Missing some important details (timing, specific people, etc.)
– ⚠️ Some parameter values are suboptimal but not wrong
– ⚠️ May include unnecessary parameters that don't harm

### INCORRECT (Score: 0.0)
Major failures in parameter selection:
– ❌ Missing most critical parameters
– ❌ Wrong people, systems, or resources specified
– ❌ Parameters would not resolve the bottleneck
– ❌ Fundamental misunderstanding of what's needed
– ❌ Parameters might make the situation worse

## Calibration Examples

### Example 1: CORRECT – Semantic Equivalence
**Bottleneck**: "Rachel needs budget approval from CFO Tom Bradley for Q4 marketing campaign by October
    1st"

**Expected Parameters**:
```json
{
  "to": ["tom.bradley@company.com"],
  "subject": "Q4 Marketing Budget Approval Request",
  "body": "Request for $50K marketing budget approval",
  "priority": "high"
}
```

**Agent Selected**:
```json
{
  "to": ["tom.bradley@company.com"],
  "subject": "Urgent: Q4 Marketing Budget – Approval Needed by Oct 1",
  "body": "Hi Tom, I need approval for the Q4 marketing budget ($50K) to proceed with the campaign.
      Deadline is October 1st.",
  "priority": "high"
}
```

**Analysis**:
– All critical elements present (recipient, urgency, amount, deadline)
– More detailed subject line improves clarity
– Body includes deadline context
– Would successfully resolve the bottleneck

**Judgment**: CORRECT

### Example 2: PARTIALLY_CORRECT – Missing Key Details
**Bottleneck**: "Project Alpha delayed because Lisa Chen hasn't reviewed technical specifications in
    JIRA ticket ALPHA-234"

**Expected Parameters**:
```json
{
  "assignee": "lisa.chen",
  "ticket_id": "ALPHA-234",
  "comment": "Hi Lisa, please review the technical specs. This is blocking Project Alpha.",
  "due_date": "2024-03-20",
  "priority": "critical"
}
```

**Agent Selected**:
```json
{
  "assignee": "lisa.chen",
  "comment": "Please review the technical specifications as soon as possible.",
  "priority": "high"
}
```

**Analysis**:
– Correct person assigned
– Missing critical ticket_id (ALPHA-234)
– No due date specified
– Priority close but not "critical"
– Generic message lacks context

**Judgment**: PARTIALLY_CORRECT – Would reach right person but lacks specificity

### Example 3: INCORRECT – Wrong Approach
**Bottleneck**: "Sales team can't access new CRM because IT hasn't completed Active Directory group
    setup"

**Expected Parameters**:
```json
{
```

```
  "ticket_type": "access_request",
  "group_name": "CRM_Sales_Users",
  "members": ["sales-team@company.com"],
  "system": "Salesforce",
  "urgency": "immediate"
}
```

**Agent Selected**:
```json
{
  "to": ["sales-team@company.com"],
  "subject": "CRM Access Information",
  "body": "The new CRM system will be available soon. Please wait for IT to complete setup."
}
```

**Analysis**:
- Completely wrong action type (email vs access request)
- Doesn't actually request the AD group setup
- Informs sales team instead of resolving with IT
- Would not resolve the bottleneck

**Judgment**: INCORRECT - Misunderstands the required action

## Parameter Evaluation Guidelines

### Consider Valid Variations
- **Email addresses**: "john@company.com" vs "John Smith <john@company.com>"
- **Dates**: "March 23, 2024" vs "2024-03-23" vs "next Friday"
- **Priority**: "high" vs "urgent" vs "critical" (if contextually similar)
- **Lists**: Order rarely matters unless sequence is critical

### Critical vs Optional Parameters
Identify which parameters are:
- **Essential**: Must be present for action to work
- **Important**: Significantly impact effectiveness
- **Optional**: Nice to have but not required
- **Contextual**: Depend on specific situation

### Common Pitfalls to Avoid
1. **Over-penalizing format differences**: JSON structure vs semantic meaning
2. **Ignoring context**: Parameters should fit the specific bottleneck
3. **Requiring exact matches**: "ASAP" vs "urgent" may be equivalent
4. **Missing parameter relationships**: Some parameters depend on others

## Output Instructions

Analyze systematically, then provide your judgment in this JSON format:

```json
{
  "judgment": "<CORRECT|PARTIALLY_CORRECT|INCORRECT>",
  "reasoning": "<2-3 sentences explaining how the parameters would or wouldn't resolve the bottleneck>",
  "parameter_analysis": {
    "critical_parameters_met": <true|false>,
    "would_resolve_bottleneck": "<yes|partially|no>",
    "missing_parameters": ["<list any critical missing params>"],
    "incorrect_parameters": ["<list any wrong params>"],
    "semantic_matches": ["<list params that match semantically>"]
  },
  "confidence": <0.0-1.0>
}
```

Remember: Focus on whether the parameters would effectively resolve the specific bottleneck in this
    context.
```

Listing 22: judge_action_parameter_scoring.j2

# D    ABLATION STUDIES

We conduct two ablation studies to validate key design decisions in our benchmark construction and assess the robustness of our evaluation framework.

## D.1    IMPACT OF CONTEXT WINDOW SIZE

Our benchmark uses 75 distractor documents per sample, selected as the maximum feasible quantity within our computational budget. To investigate whether increased context length fundamentally increases task difficulty, we generated 100 samples for each distractor quantity $k \in \{50, 75, 100\}$ and evaluated model performance across these configurations.

Figure 4 demonstrates a clear performance degradation as context size increases. At the baseline of 50 distractors, GPT-5, Claude Opus 4.1, and Claude Sonnet 4 achieve comparable performance. However, this gap widens significantly as the number of distractors grows to 100, where GPT-5 maintains 0.377 F1 while Claude models drop to 0.293 (Opus) and 0.256 (Sonnet) respectively.

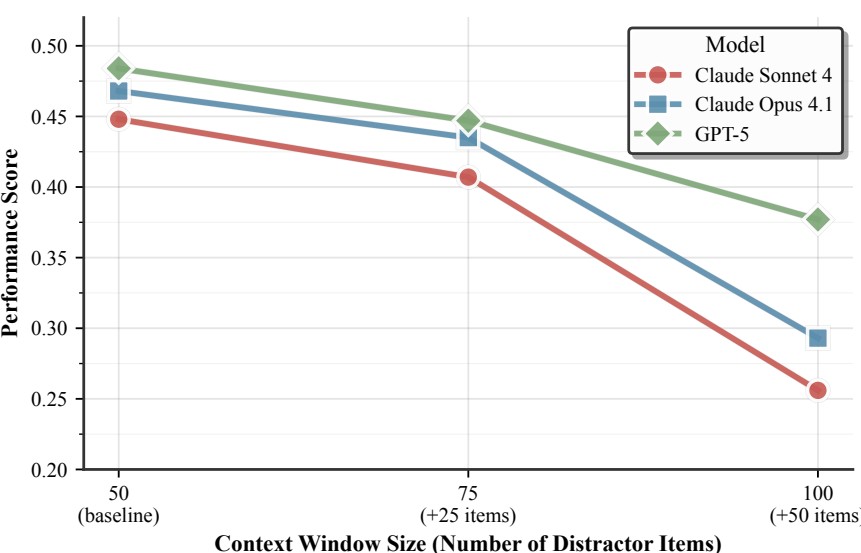

Figure 4: Performance degradation with increasing context window size. All models show declining performance as distractor count increases from 50 (baseline) to 100 (+50), with Claude models exhibiting steeper decay than GPT-5. This demonstrates the fundamental challenge of maintaining search and reasoning accuracy in long context, high-distractor environments.

This differential degradation reveals distinct capabilities in long-context reasoning among frontier models. While all models suffer from increased distractor count, Claude models exhibit significantly steeper performance decay. These results also suggest that bottleneck identification and resolution become increasingly difficult as agents must reason over and eliminate larger volumes of contextually plausible but ultimately irrelevant information.

## D.2 DATA GENERATION MODEL DIVERSITY

To mitigate the risk of model-specific artifacts that could artificially inflate performance for models from the same family, our full benchmark employs three distinct LLMs in the data generation pipeline. To quantify the impact of this design choice, we conducted a controlled study where we generated 53 samples (with 75 distractors each) using three different data generation models independently: GPT-5-mini, GPT-4.1, and Claude Sonnet 4. We then evaluated GPT-5's performance on each dataset variant. Note that this represents a simplified setting compared to our pipeline, which combines multiple models within each sample to create more complex artifacts.

Figure 5 reveals substantial cross-family performance disparities. GPT-5 achieves strong results on data generated by models within its own family (0.951 retrieval F1 on GPT-5-mini data, 0.882 on GPT-4.1 data), but performance drops steeply on Claude Sonnet 4-generated data (0.564 retrieval F1). This pattern persists across all evaluation dimensions: end-to-end success rates reach 30.2% and 51.9% for GPT-family generated data but fall to just 26.4% for Claude-generated samples.

These results motivate our multi-model data generation strategy. The 25.5-point difference in end-to-end performance between GPT-4.1 and Claude Sonnet 4 data indicates that single-model generation can imprint family-specific artifacts, enabling pattern-based shortcuts rather than genuine reasoning. By mixing data from diverse model families, our benchmark better tests general reasoning ability instead of overfitting to a single model's heuristics.

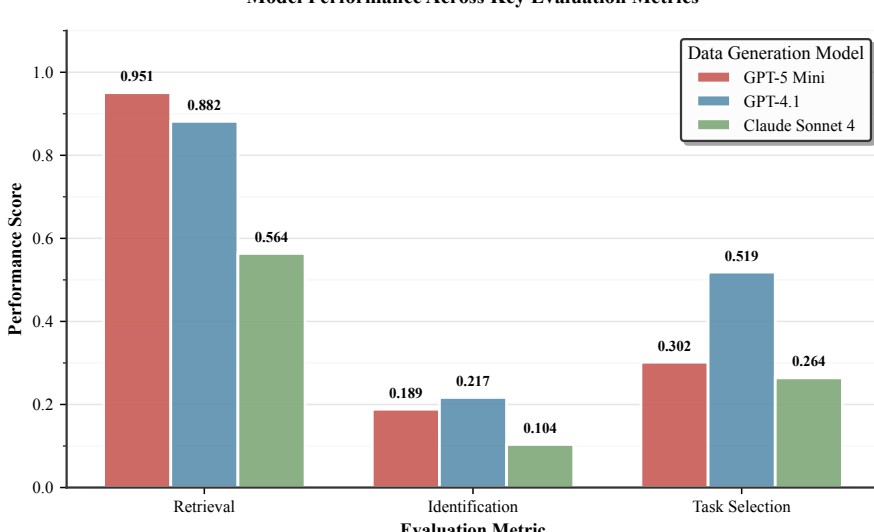

Figure 5: GPT-5 performance on data generated by different models. GPT-5 exhibits much higher search performance on samples generated by its own model family (GPT-5-mini: 0.951 F1, GPT-4.1: 0.882 F1) compared to cross-family generation (Claude Sonnet 4: 0.564 F1), validating our multi-model data generation approach.

## E  NOTE ON THE USE OF LANGUAGE MODELS

We utilized Claude (Anthropic) as an AI writing assistant throughout the preparation of this manuscript. Claude was employed primarily for refining sentence clarity, improving paragraph flow, and ensuring consistency in academic writing style. All scientific content, experimental design, analysis, and intellectual contributions remain solely those of the authors.

