# OpenReview forum: "Beyond Reactivity: Measuring Proactive Problem Solving in LLM Agents}"
_ICLR.cc/2026/Conference — Submitted to ICLR 2026_

### Official Review · Reviewer_9VNE · 2025-10-16

**Soundness:** 3
**Presentation:** 2
**Contribution:** 2
**Rating:** 4
**Confidence:** 4

**Summary:**

This paper introduces a benchmark for evaluating proactivity of LLM agents. The authors define proactivity as a three-stage process: 1) searching over datastore; 2) identifying the most critical bottleneck; 3) executing an appropriate resolution.
They construct a synthetic data generation pipeline, which consists of a  synthetic world model to provide persona profile, and a user datastore to simulate the working environment context. They created 1,000 diverse test samples, and test LLMs like GPT-5, Claude Opus-4.1 and agent frameworks like ReACT, Reflexion, ReWoo, showing that current models performs poorly on the benchmark.

**Strengths:**

1. The paper addresses an important and under-explored problem. Proactivity is an important aspect intelligent agents, yet current research mainly focuses on reactive paradigms. Constructing a benchmark for this direction is valuable.
2. The proactive behavior is decomposed into  three distinct stages: search, identification, and execution. This provides a clear and interpretable lens for analyses and could inspire more principled model designs.

**Weaknesses:**

1. The paper shares similar concepts with Lu et al. (2024), which also investigates proactive behavior and synthetic benchmarks. It would be benefit for the paper to clearer explain how it extends or complements that prior work.
2. Unlike Lu et al. (2024), which also proposed models and data pipelines to improve proactivity, this work remains only an evaluation benchmark, without algorithmic insights or new modeling strategies.
3. The benchmark scenarios are restricted to synthetic, office-style contexts. It is unclear whether these setups can generalize to other forms of proactive tasks such as embodied or multimodal environments.
4. The environment information is under-specified. The environment (or user datastore) is central to enabling proactivity, because agents must perceive the current state and generate appropriate proposals. More details and explanation are needed to show how much genuine proactivity can be tested under this setup.
5. The submission appears somewhat rough:
    * The title on the ICLR submission page includes an extra closing brace “}”;
    * Several lines contain garbled symbols (e.g., lines 92, 296, 364, 560, 566, 1064, etc.);
    * The references list the same Lu et al. paper multiple times as “2024a/2024b.”
The authors can carefully polish and refine the paper to improve its overall clarity and presentation quality.

**Questions:**

Please refer to the Weaknesses above.

---

> ### Author Response · Authors · 2025-11-24
>
> We greatly appreciate your feedback - it was used to significantly improve our submission in revision. We address each weakness below:
>
> (1) Relation to Lu et al. (2024)
>
>  We understand this point and appreciate the opportunity to clarify this. Lu et al. introduced the idea of proactive agents, a reward model of users for evaluation and present the ProactiveBench dataset. Their task is based on predicting proactive assistance from the current activity context (events and state).
> PROBE explicitly extends this line of work by adding search over long context, multi-document user datastores as a task and a significant addition of complexity. It also operationalizes bottleneck identification across dispersed evidence and tests parameterized task execution as separate, scored stages. We believe these to be important extensions, as knowledge workers often struggle with context distributed over a large quantity of sources.
> To address the weakness directly, we added a paragraph to the Related Works (Agent Benchmarking Section, paragraph 2) directly addressing how we extend Lu et al.
>
> (2) Benchmark and Algorithmic insights
>
> We appreciate the opportunity to address this. PROBE is designed as a model-agnostic evaluation benchmark, akin to SWE-bench for coding, so that future work can plug in arbitrary architectures. Beyond the novel framing of the benchmark as a long-horizon workplace proactivity measurement, we also contribute:
>  (i) A scalable pipeline that can be used to easily change the context size and difficulty of our benchmark by merely tweaking parameters, so that the benchmark can scale to real workplace scenarios of varying contextual difficulty.
>  (ii) The decomposition of long-horizon proactivity into search → bottleneck identification → task execution
> (iii) systematic evaluations and failure-mode analyses of frontier LLMs and agentic frameworks on this pipeline, which already surface concrete algorithmic weaknesses (e.g., root-cause and interpersonal reasoning).
>
> Surfacing such weaknesses, while not equivalent to algorithmic development, can be used to surface concrete axes of improvement (Root Cause Analysis, Parameter Filling, Attribution) and thus aids development going forward. In the revised Related Works, we’ve more clearly highlighted our holistic impact.
>
> (3) Synthetic scenarios, generality, and environment specification
>
> We thank the reviewer for the constructive comment; multimodality is an excellent idea for extending PROBE datastores. Our initial domain choice is intentionally “office-style” (email, calendar, documents), because this is where proactive digital agents are currently most deployable. It is also an area where long-horizon document reasoning is both natural and high-impact. However, the underlying formulation (world model, bottlenecks, document set, and parameterized action space (A, P)) is modality-agnostic and can be extended to embodied or multimodal settings in future work. We note that our contribution of long-horizon + long-context datastores can benefit these settings significantly, as temporal context from an embodied agent’s conversations can always be stored in a user datastore as a document.
> As for the environment, we note that our world models and datastores - which are a representation of one’s environment - are quite comprehensive. A typical world model will detail a user’s workplace hierarchy, relationships, communication preferences, goals, and more - providing significant context that an agent can use to display genuine proactivity.  We believe this environment construction can generalize across a large number of proactive contexts and information types. To address this concern directly, we’ve added a World Model from the benchmark in Appendix B, showing a detailed specification of user state that a workplace agent could use to take proactive action.
>
> (4) Presentation roughness (title brace, garbled symbols, duplicate references)
>
>  We greatly appreciate the reviewer’s thoroughness and the pointer to these presentation issues. We have fixed all presentation issues and cleaned up the submission significantly in our revision.

---

> > ### Comment · Reviewer_9VNE · 2025-11-24
> >
> > Thank you for the detailed response. However, the coverage and realism of the benchmark are still limited even within the text modality in terms of task types, domains, and scenarios. I remain concerned about its ability to represent broader real-world proactive agent tasks. For this reason, I will keep my original rating.

---

> > > ### Author Response · Authors · 2025-11-26
> > >
> > > We're grateful for your continued engagement and for the opportunity to address these concerns.
> > >
> > > We'd welcome any feedback as to which task types, domains, or scenarios are being referred to. This would either help us detail how our current benchmark addresses these scenarios or identify a concrete area for improvement.
> > >
> > > That said, we'd also like to note that many important benchmarks achieve impact through deep coverage of a particular domain as opposed to a broad coverage of many: SWE Bench focuses exclusively on Python repositories, MATH exclusively on competition math, and GSM8K exclusively on grade school math. Hence, we consider the depth-first approach of the benchmark a feature rather than a bug - office style proactivity represents an enormous deployment opportunity affecting many millions of knowledge workers, and evaluation in this space is a genuine need.
> > >
> > > To address your concerns more concretely, we can add 100-200 samples in the camera-ready version with additional representations of multimodal document types such as "voice transcripts" and "image captions" to address the extension of the benchmark to multimodal scenarios, if this would meaningfully improve the contribution.
> > >
> > > We hope this addresses concerns about realism; we are committed to ensure the benchmarks coverage of real world office-style proactive scenarios to demonstrate the extensions of our datastore-centric design.

---

### Official Review · Reviewer_UZ3r · 2025-10-28

**Soundness:** 1
**Presentation:** 2
**Contribution:** 1
**Rating:** 2
**Confidence:** 3

**Summary:**

The authors propose PROBE, a benchmark designed to evaluate the capabilities of proactive LLM agents in searching, identifying issues, and executing actions to solve them. They employ a world model to generate the entire dataset, which contains 1,000 entries, and have three annotators assess the difficulty of the generated samples. Among these, only 26 samples were successfully completed by humans. The authors further compare the performance of several state-of-the-art models, highlighting the challenging nature of the dataset. In addition, they conduct an error analysis to identify and categorize the key issues faced by proactive agents.

**Strengths:**

1. The authors present a clear and systematic formulation of the data generation pipeline, which helps readers understand the overall design and underlying methodology of the proposed benchmark.
2. The paper provides a detailed error analysis, enabling readers to identify the key bottlenecks and challenges in proactive problem-solving for LLM-based agents.

**Weaknesses:**

1. Lack of practical validation for the benchmark design. There is no clear evidence demonstrating that the constructed benchmark is well designed from practical or realistic considerations. It is particularly concerning that even human experts failed to complete most of the tasks, while some state-of-the-art models managed to perform better — a result that appears counterintuitive and warrants further explanation.
2. Unreliable evaluation using LLM-as-a-Judge. The evaluation process lacks robustness, as the authors only assessed 50 GPT-4.1 prediction–output pairs, with no detailed description of the evaluation criteria, sampling process, or inter-rater consistency. This raises doubts about the reliability and reproducibility of the reported results, especially when considering the human experts failed in Table 2.
3. Unfair comparison in the agentic framework. The proposed agentic framework demonstrates very poor performance, which raises concerns about the fairness of the comparisons made. As mentioned by the authors (line 387), the current setup may not ensure an equitable evaluation across different models. Additional experiments or ablation studies are needed to provide a more balanced and fair comparison.

**Questions:**

1. Could the authors provide more details about the human evaluation process? Specifically, how is task difficulty validated when human annotators achieve near-zero performance in the Bottleneck Identification component?
2. Is the benchmark entirely generated by language models? If so, which model(s) were used, and what was the rationale behind their selection?
3. How do the authors ensure that the evaluation on PROBE accurately reflects real-world agent proactiveness? Does the simulation include any real-world scenarios or instances, or is it purely synthetic? Clarifying this would help assess the benchmark’s external validity.

---

> ### Author Response · Authors · 2025-11-24
>
> Thank you for your feedback - it was used to greatly improve the quality of the paper in our revision. We respond to weaknesses and questions below:
>
> Practical validation
>
> We value the reviewer’s concern about practical validity and agree that this needs to be clarified. We address some ambiguity over realism with qualitative evidence: a concrete world model + bottleneck example in Appendix B, and a categorization of bottleneck types in Figure 3.
>
> To show that the reason LLMs outperformed humans had more to do with long context size than poor task formation, we performed additional human evaluations to test how humans would fare when context was shortened. In these experiments, we reduce context size by removing distractors and summarizing true positives. We note that this is still not a trivial task: task selection remains a 26 choice multiple choice question and identification is a free response requiring dense reasoning across 1000+ tokens. The study is found in Section 2.5.
>
> We see that the challenge with the dataset is not the quality, but primarily the amount of context that annotators have to keep in their mind as part of solving the task, as evidenced by the drastic improvement in human scores upon reducing context size. Language Models, on the other hand, suffer less from context limitations.
>
>
> LLM-as-judge reliability
>
> We agree with the reviewer’s emphasis that evaluation robustness is important and appreciate the opportunity to expand upon our initial description. Using LLMs as judges is now a standard and extensively studied practice: recent surveys and benchmarks (e.g., A Survey on LLM-as-a-Judge [1]; MT-Bench / Chatbot Arena [2]) show that strong LLM judges can reach human–human levels of agreement (>80%) when carefully prompted. In our case, we use the strongest model available to us (GPT-5) as the evaluator, observe high inter-rater consistency (Section 3.2), provide our structured evaluation criteria prompt (Appendix C.6). We observe high agreement consistent with prior work. To address this concern, we've modified Section 3.2 to include the sampling process and inter-annotator agreement, and add a reference to the evaluation criteria in Section 3.1.
>
> [1] A Survey on LLM-as-a-Judge. Gu et al, 2024
> [2] Judging LLM-as-a-Judge with MT-Bench and Chatbot Arena. Zheng et al, 2024
>
> Fairness of agentic comparisons
>
> We apologize for not making this point sufficiently clear. We’ve rephrased Section 3.5 (paragraph 2) to better clarify and emphasize why comparisons are fair.
>
> Q1: human evaluation process
>
> As clarified in Sec. 2.4 / Table 2, the human evaluation was designed to (i) validate instance realism and (ii) quantify how difficult the task is when framed as a benchmark. Human annotators, given the same instructions as models, reported that the documents and actions were realistic and the tasks were understandable, but required substantial effort. The “near-zero” bottleneck score arises from the very-long-context nature of the benchmark: reasoning over as many as 122,000 tokens is both realistic as an LLM task and extremely difficult for any human being. To address this question, we’ve included section 2.5 in the revision, validating that most of the task difficulty stemmed from the long-context nature of the data.
>
> Q2. Models used and rationale
>
> The benchmark datastores are generated with GPT-4.1 as the primary generation model (Section 2.4), using the process described in our methodology section (Section 2.3).
> We chose GPT-4.1 because it is the strongest model we could feasibly use at scale (considering generation speed, cost, and rate limits), given that each instance averages ~100k tokens, while using GPT-5 (or Claude-Opus-4.1) for generation would have been prohibitively slow (GPT-5) or expensive (Claude-Opus-4.1).
>
> Q3. Real-world realism and external validity
>
> [clarified with reviewer ZmBb as well]
> In our setting, using real user data (long-horizon email, calendar, internal docs) would require sharing highly sensitive personal logs, which is not feasible for a public benchmark, so PROBE intentionally targets synthetic but structurally realistic datastores. Concretely, we use real LinkedIn persona information to contextualize all stages of the generation pipeline. Our final check for realism - the feedback we received from human annotators who were independent of the data generation process - was that documents and bottlenecks seemed realistic.
> This design follows the same spirit as prior widely used benchmarks that fully simulate users and documents via templates like Appworld [1]
>
> [1] AppWorld: A Controllable World of Apps and People for Benchmarking Interactive Coding Agents. Trivedi et al. ACL 2024 (Best Resource Paper)
>
> To address your concern more directly, we’ve (1) Added a generated World Model and bottleneck in Appendix B to qualitatively show the realism of generations and (2) Included Figure 3, which presents the tangible workplace scenarios which the bottlenecks fit into.

---

> > ### Comment · Reviewer_UZ3r · 2025-11-25
> >
> > Thanks for your detailed response — you have addressed my concerns regarding W2 and Q1&2, and I will raise my score accordingly. However, I still have concerns about W1&3 and Q3. The experimental section should be redesigned to ensure that the results can be validated with reliable metrics rather than relying solely on LLMs.

---

> ### Author Response · Authors · 2025-11-26
>
> We appreciate your continued engagement and feedback, and would like to help address the remaining weaknesses and questions you have raised:
>
> - Weakness 1: Practical validation for the benchmark design + Concerning that humans failed in task
>
> We appreciate this point, and have taken three steps in revisions to address it.
>
> 1) In our most recent revision, we increased the sample size for our LLM-as-a-judge human annotation study in Section 3.2 (now 100 data points each), and continue to observe a high inter-annotator agreement (85%).
>
> 2) Task Clarity / Data Quality in Section 2.5 (Table 3), which we added in the last revision, provides evidence of human ability to understand and perform the task. When reducing the task context size to human legible amounts in this table, we also observe reasonable performance and a substantial Fleiss' Kappa (0.714), presenting practical validation that the task is well framed, doable, and legible to humans.
>
> 3) As qualitative evidence, we've also added a world model and bottleneck example in Appendix B.
>
> - Weakness 3: Fairness of Agentic comparisons
>
> We appreciate the chance to clarify this point. Our original intent in line 387 was not to suggest that the evaluation is inequitable, and we have rephrased it in the revision. The original goal was to acknowledge that many popular agent frameworks are typically demonstrated with large tool suites (web search, APIs, etc.) which do not help in the instance of the skill we benchmark.  In PROBE experiments, we standardized the setup on a small, shared tool set over the user datastore to ensure equitable comparison.
>
> All models and agent frameworks in our experiments operate under this same constrained interface, so comparisons within PROBE are fair; the lower performance of off-the-shelf frameworks reflects their difficulty in adapting to a constrained, datastore-centric proactive setting, not that they were deprived of tools that other models had. We've re-adapted Section 3.5 (paragraph 2) in our latest revision to clarify this explicitly.
>
> - Question 3: How do the authors ensure realism?
>
> We appreciate the opportunity to better explain this.
>
> 1) We leverage real LinkedIn profiles at every stage of the data generation pipeline, so all bottlenecks and world modelsin the benchmark are contextualized by real persona descriptions.
>
> 2) We provide the opportunity to verify realism qualitatively in Appendix B
>
> 3) We show the realistic workplace scenarios that bottlenecks fit into in Figure 3.
>
> In our setting, sharing real user data (emails, calendar events, documents) would be a cause major privacy issues, so PROBE uses synthetic - but structurally realistic - datastore. In this practice, we follow precedents such as Appworld.
>
> - Final concern: Solely relying on LLMs
>
> We apologize for the confusion and appreciate the opportunity to restate this more clearly.
>
> We solely rely on LLMs for only a single intermediate metric (Identification) whereas Search metrics are entirely LLM-free, and Task-Execution does not rely on LLMs for Task Selection scoring.
>
> To further address this concern, in the camera-ready version we will also report the end to end Task Selection metric, which will be an end-to-end metric for our task that is computed entirely without LLMs (computed as an exact match).

---

### Official Review · Reviewer_gyiy · 2025-11-01

**Soundness:** 3
**Presentation:** 3
**Contribution:** 3
**Rating:** 6
**Confidence:** 4

**Summary:**

This paper introduces a benchmark, PROBE, to test LLMs abilities at proactively identifying issues. PROBE is synthetically generated using personas extracted from real LinkedIn users and an LLM generated bottleneck from which problematic and distractor documents are generated to be identified. PROBE tests LLMs abilities at searching for problematic documents, identifying bottlenecks, and choosing the right solution from a set of actions to address the bottleneck.

**Strengths:**

$Originality$: This paper introduces a new benchmark testing LLMs ability at proactiveness in a new setting: identifying bottlenecks, problematic documents, and resolution actions given a world model (user)

$Quality$: The benchmark has been tested carefully including a user study that revealed the difficulty of the tasks and an LLM-as-a-judge evaluator that is well aligned with annotators

$Clarity$: The benchmark details were easy to understand and the results are clear

$Significance$: Proactive LLMs is an increasingly important field that requires more benchmarks

**Weaknesses:**

- The user study was conducted on about 3% of the entire benchmark and performance was very low which could suggest the benchmark is difficult or the synthetic generation pipeline has issues
- The quality of the benchmark is based on iterating over prompts and running an adversarial agent to locate artifacts in 5 samples until no artifacts are detected.

**Questions:**

- Why does the Bottleneck Identification performance plateau around 42-43% across various models?
- How does GPT-4.1-mini achieve the same bottleneck identification performance as GPT-4.1 with significantly worse search ability?
- The agent frameworks are very low compared to the comparable base model GPT-4.1-mini (which the caption states is similar to GPT-5 mini performance). This is surprising to me, especially since Reflexion performs about the same as ReAct. Where are the baseline / base model prompts for conducting the action located and where do they differ?

---

> ### Author Response · Authors · 2025-11-24
>
> Thank you for your review - you raised strong points that were used to significantly improve the paper in our revision. We'll address weaknesses and questions one by one:
>
> -User study:
> We appreciate this point, and we agree that human validation of a synthetic benchmark is essential. Our user study was scoped as a sanity check rather than a re-annotation of the dataset: even 3% of PROBE is costly to evaluate because each instance spans 70-80 long documents and requires multi-step reasoning. In that subset, humans consistently judged the scenarios as realistic and understandable but genuinely hard due to context size.
>
> To address this concern and provide evidence that the difficulty of the task stems from context size as opposed to generation pipeline issue, we’ve performed additional human evals to validate data quality. In these experiments, we reduce context size by removing distractors and summarizing true positives. We note that this is still not a trivial task, as task selection remains a 26 choice multiple choice question and identification is a free response requiring extensive cross document reasoning across 1000+ tokens. The study is found in Section 2.5.
>
> We see that the challenge with the dataset is not the quality, but primarily the amount of context that annotators have to keep in their mind as part of solving the task, as evidenced by the drastic improvement in scores upon reducing context size.
>
> -5-sample validation:
> Thanks for pointing this out - we really appreciate the opportunity to clarify this point. In section 2.4, the “5 samples” are not meant as a claim that we fully validate the entire benchmark on 5 instances, but as an additional prompt-level stress test on top of other validation: we use an adversarial agent on a small stress set to hunt for global artifacts (e.g., templatic shortcuts, positional cues) that would generalize to the whole distribution. When it finds a shortcut on those examples, we change the generation prompts, which in turn affects all later data generation. This adversarial loop is complemented by human inspection of ~20-30 samples and now also our data quality study in section 2.5.
>
> Questions:
>
> Q1: Why does Bottleneck Identification plateau at ~42-43%?
>
> Our error analysis (Table 5, Section 4) reveals that root cause identification represents the dominant failure mode, accounting for ~74% of identification errors across all models. This suggests the plateau reflects a fundamental reasoning challenge (bottlenecks are found across multiple long documents and require ignoring several non-trivial distractors) rather than a model-specific limitation.
>
> Q2: How does GPT-4.1-mini achieve comparable Bottleneck Identification with inferior Search performance?
> This phenomenon, which we term "shortcutting" (Section 3.3), occurs when models leverage stronger free-form reasoning to compensate for incomplete evidence retrieval. However, this creates a critical gap: being "right for the wrong reasons" does not translate to executable solutions. GPT-4.1-mini's Task Execution score (0.20) is half that of GPT-4.1 (0.38), confirming that faithful evidence use is essential for end-to-end success.
>
> Q3: Why do agent frameworks underperform base models?
>
> Two factors explain this gap:
>
> (1) Constrained tool access: Our experimental setup restricts agents to SQL and semantic search, while production agents typically leverage web search and APIs. This limits their core "act" channel for iterative knowledge acquisition: we’ve clarified this in the updated Section 3.5.
>
> (2) Cascading errors: Poor retrieval (F1: 0.12-0.25 vs. 0.46-0.65 for base models) propagates through the pipeline, preventing accurate bottleneck identification regardless of reasoning quality.  This highlights that retrieval remains the critical bottleneck for agentic systems, in particular dense retrieval that requires cross document reasoning. We will release all agentic prompts along with the code and camera-ready version of the paper.
>
> As for the Reflexion vs. ReAct comparison: In the search stage of PROBE, reflection on a search query is not particularly helpful due to the fact that the action space (the sheer amount of queries one could run) is essentially infinite. As a result, Reflexion fails to significantly improve search, and the search errors naturally cascade down the PROBE pipeline.

---

### Official Review · Reviewer_ZmBb · 2025-11-02

**Soundness:** 2
**Presentation:** 2
**Contribution:** 2
**Rating:** 4
**Confidence:** 3

**Summary:**

This paper introduces PROBE, a benchmark and accompanying pipeline designed to evaluate the proactivity of LLMs through a structured workflow: searching unsolved issues, identifying bottlenecks and executing appropriate resolutions. The benchmark has been applied across diverse models and agentic frameworks, with results revealing significant challenges in achieving effective proactive assistance.

**Strengths:**

1. This paper introduces a focus on long-term proactivity, which meaningfully distinguishes it from prior work on proactive tasks that typically address short-horizon or immediate user intents.
2. The paper presents a well-structured and systematic pipeline for constructing the PROBE benchmark—a notable contribution that advances the methodology for synthetic data generation in agent evaluation.

**Weaknesses:**

1. Insufficient Benchmark Description: The paper does not adequately present the PROBE benchmark. It only reports aggregate statistics—such as the number of tokens, actions, and documents without providing concrete details about the tools used, the scenarios covered, or other essential contextual information.
2. Unclear Realism Gap: The distinction between synthetic data and real-world scenarios remains ambiguous. The claim that the benchmark is “realistic” appears largely subjective. In contrast, works like *Proactive Agent* using real-world data in their evaluation.

**Questions:**

1. What specific scenarios does your benchmark include? This information is essential for evaluating the benchmark’s scope and should be clearly presented in the main paper.
2. What distinguishes your benchmark from previous benchmarks?
3. When and how will your agent perform proactive actions? How could users or annotators acknowledge their need for proactive problem solving in your settings?
4. Appendix Section B.3.1 shows the prompt for search, identification, and task-selection pipeline. The current design requires the model to output all three actions in a single response. However, this should be a multi-step process. How could LLMs leverage the context of searching results?
5. How does persona meaningfully influence LLM behavior or performance? From the current description, personas are used primarily to structure interpersonal relationships and contextualize bottlenecks. Yet, since all relevant evidence (e.g., emails, documents, calendar entries) is explicitly provided, it remains unclear how “person attribution” affects bottleneck identification.

---

> ### Author Response · Authors · 2025-11-24
>
> Thank you for your review - it was used to significantly improve the quality of the paper in the revision. We’ll address the points one by one:
>
> - Insufficient Benchmark Description:
> Thank you for the feedback, and we  acknowledge that  - while we did our best to describe the benchmark methodology, we could have been more explicit about certain implementation details. To address this, we have expanded the benchmark description, added a figure showing the bottleneck categories, and added an example of an actual generated world model and bottleneck in the appendix. All updates can be found in or are linked by the updated section 2.4.
>
> - Unclear Realism Gap:
> Thank you for raising this. In our setting, using real user data (long-horizon email, calendar, internal docs) would require sharing highly sensitive personal logs, which is not feasible for a public benchmark, so PROBE targets synthetic but structurally realistic datastores. Concretely, we generate and filter documents with LLMs plus human checks so that both the artifacts and the required actions qualitatively resemble genuine workplace behavior. Our final check for realism - the feedback we received from human annotators who were independent of the data generation process - was that documents and bottlenecks seemed realistic.
> This design follows the same spirit as prior widely used benchmarks that fully simulate realistic users and documents via templates like Appworld [1]
>
> [1] AppWorld: A Controllable World of Apps and People for Benchmarking Interactive Coding Agents. Trivedi et al. ACL 2024 (Best Resource Paper)
>
> - To address your concern more directly, we’ve (1) clearly stated that PROBE uses synthetic user datastores, (2) added a generated World Model and bottleneck in Appendix B to qualitatively show the realism of generations,  and (3) Included Figure 3, which presents the tangible workplace scenarios which the bottlenecks fit into.
>
>
> Questions:
>
> (1) Scenarios and scope:
>
> Thank you for pointing this out. The current draft does not surface the scenario taxonomy clearly enough. We’ve now included Figure 3 in the paper to clarify the included Bottleneck scenarios. The other component of our benchmark which falls into a partition are our document types: Calendar Events, Emails, and Documents (2.3, 177-182).
>
> (2) What distinguishes PROBE from prior benchmarks:
>
> We appreciate the opportunity to clarify this. Current proactive benchmarks are often benchmarked on their ability to make suggestions on the basis of very recent temporal context, failing to capture proactive insights that emerge through longer-term analysis. This requires discovering problems that are not explicitly mentioned in the current turn, but only implicit in the evolving state of the user’s work. We have made this distinction more explicit in the Related Work (Agentic Benchmarking Section) by contrasting PROBE’s “search → bottleneck identification →  resolution” loop with prior benchmarks that primarily test task execution from a given prompt.
>
> (3) When/how proactive actions happen & how “need” is defined"
>
> Thank you for the question! PROBE is designed to approximate an always-on assistant that regularly inspects a user’s evolving datastore. The need for proactive problem solving is encoded in the ground truth: for each scenario, the annotation  specifies which bottlenecks truly matter for the user’s goals and what constitutes an acceptable resolution.   We hope to have clarified this by providing an example of a proactive bottleneck scenario in Appendix B, and the inclusion of proactive scenarios in Table 3.
>
> (4) Single-shot vs multi-step search–identify–act:
>
> We agree with you that, in a realistic deployment, search, identification, and task selection are naturally multi-step, and LLMs should be able to condition on search results iteratively. The prompt shown in Appendix C.3.1 is only the evaluation criteria; the checklist defines the ground truth by which we compare models. That is: after an agent outputs the search, identification, and selected action through, we use the checklist just to score its responses. In other words, PROBE supports multi-step pipelines that leverage search results; the appendix prompt  C.3.1 is simply the evaluation criteria. We have updated the prompt in C.3.1 to explicitly denote this.
>
> (5) Role of personas in behavior and evaluation:
>
> We appreciate the concern raised. Personas are used to create the social context, priorities, and responsibilities in the synthetic workplace: who depends on whom, which relationships are high-stakes, and what constraints (workload, communication style, preferences) the user operates under. This, in turn, affects which emails, calendar events, and documents are generated (e.g., a PM vs. an IC engineer receives very different messages/deadlines). We acknowledge that this should have been better spelled out,  and have updated Section 2.3 (177-182) and included Appendix B to better convey this personalization.

---

### Meta-Review · Area_Chair_k8Th · 2026-01-05

**Summary:**

**1) Summary**
This paper introduces PROBE, a benchmark designed to evaluate LLMs’ proactive capabilities across three stages: searching for issues, identifying bottlenecks, and executing appropriate resolutions. The benchmark is built through a synthetic pipeline that generates personas, documents, and bottlenecks, and is used to test a range of models and agentic frameworks. Results show that current systems perform poorly, suggesting that long-horizon proactive reasoning is still an open challenge.

**2) Strengths**

* Proactivity is framed as a multi-stage process (search, identification, execution), offering a structured and interpretable evaluation framework.
* The benchmark highlights an under-explored yet important capability of intelligent agents: long-horizon proactivity beyond short-term reactive tasks.
* The synthetic data generation pipeline is systematically presented, enabling a consistent methodology for building proactive evaluation scenarios.

**3) Weaknesses**

* Limited benchmark specification: toolsets, scenario types, and environment details are under-described, leaving unclear what forms of proactivity are actually measured.
* Realism and validity concerns: reliance on synthetic office-style scenarios, minimal human validation, and human underperformance raise questions about benchmark practicality and design soundness.
* Evaluation robustness is weak due to sparse and insufficiently documented LLM-as-judge assessments and lack of inter-rater checks.
* Some experimental comparisons appear unfair or inconclusive, with agent frameworks underperforming without adequate justification or ablations.
* Presentation issues—including roughness in text, missing details, and inconsistencies—reduce clarity and make it harder to assess the contribution.

**Reviewer Concerns:**

Two out of the four reviewers followed up with the rebuttal and expressed remaining concerns.

**Reviewer Scores:**

The reviews were generally leaning toward rejection, and none of the reviewers seemed to be willing to change their scores.

---

### Decision · Program_Chairs · 2026-01-26

Reject